**Investigation**

# Electric shock causes a fleeing-like persistent behavioral response in the nematode *Caenorhabditis elegans*

Ling Fei Tee,[1] Jared J. Young,[2] Keisuke Maruyama,[1] Sota Kimura,[1] Ryoga Suzuki,[1] Yuto Endo,[1,3] Koutarou D. Kimura[1,3,*]

[1]Graduate School of Science, Nagoya City University, Nagoya 467-8501, Japan
[2]Mills College at Northeastern University, Oakland, CA 94613, USA
[3]Department of Biological Sciences, Graduate School of Science, Osaka University, Toyonaka, Osaka 560-0043, Japan

*Corresponding author. Graduate School of Science, Nagoya City University, Nagoya 467-8501, Japan Email: kokimura@nsc.nagoya-cu.ac.jp

Behavioral persistency reflects internal brain states, which are the foundations of multiple brain functions. However, experimental paradigms enabling genetic analyses of behavioral persistency and its associated brain functions have been limited. Here, we report novel persistent behavioral responses caused by electric stimuli in the nematode *Caenorhabditis elegans*. When the animals on bacterial food are stimulated by alternating current, their movement speed suddenly increases 2- to 3-fold, persisting for more than 1 minute even after a 5-second stimulation. Genetic analyses reveal that voltage-gated channels in the neurons are required for the response, possibly as the sensors, and neuropeptide signaling regulates the duration of the persistent response. Additional behavioral analyses implicate that the animal's response to electric shock is scalable and has a negative valence. These properties, along with persistence, have been recently regarded as essential features of emotion, suggesting that *C. elegans* response to electric shock may reflect a form of emotion, akin to fear.

**Keywords:** persistency; behavioral response; voltage-gated channels; neuropeptides

## Introduction

Animal behaviors, such as feeding, mating, aggression, and sleeping, are strongly related to internal states in the brain, namely motivation, arousal, drive, and emotion (Berridge 2004; Kennedy et al. 2014; Anderson 2016). Because animals can produce different behavioral responses to the same stimulus depending on their brain state, these states are considered to be the foundation from which a variety of behavioral responses emerge (Maimon 2011; Chen and Hong 2018). The brain states persist for a certain period of time and transit to a different state based on internal and/or external triggers, which can be observed as transitions among different persistent behavioral states. The neural mechanisms of brain/behavioral states are starting to be revealed: For example, the behavioral states of mating and aggressiveness have been shown to be controlled by relatively small circuits in mice and flies (Lee and Dan 2012; Hoopfer et al. 2015; Anderson 2016). However, the mechanisms of persistent brain/behavioral states have been revealed in only limited studies, and, moreover, the molecular basis that generates persistent states is still unclear.

The nematode *Caenorhabditis elegans* has been widely used in neurobiological research because of the feasibility of molecular, physiological, and behavioral analyses of neural functions (de Bono and Maricq 2005; Bargmann 2006; Sasakura and Mori 2013). Recently, persistent behavioral states have also been studied in these animals, especially roaming/dwelling and sleep/arousal. Roaming/dwelling are states of locomotion on bacterial food that involve either moving over long distances at a constant speed or moving back and forth over short distances (Fujiwara et al. 2002; Ben Arous et al. 2009). Sleep in *C. elegans* is a phenomenon observed just before molt, and meets the definition of sleep in higher animals such as humans, rodents, fishes and flies (Raizen et al. 2008). Both the neural circuits and genes that control these phenomena are being revealed (Flavell et al. 2020). However, much remains unknown about *C. elegans*' behavioral states.

In this study, we report that *C. elegans* exhibits a novel type of persistent behavioral response to electric stimulus. The animals respond to alternating current (AC) stimulus by immediately increasing their speed, and the speed increase persists for minutes even when an electric stimulus as short as 5 seconds is provided: This result suggests that the response is caused not by direct stimulation of the motor system for rapid movement but by persistent activity of a specific set of neurons to generate the behavioral response. Further behavioral analyses suggest that the speed increase to AC stimulus is scalable and has negative valence. Because persistent behavioral response is one of the most prominent characteristics of emotions of animals (Nettle and Bateson 2012; Anderson and Adolphs 2014; Perry and Baciadonna 2017; Paul and Mendl 2018; Abbott 2020), and persistency, scalability, and valence are 3 of the 4 key features of animal emotions proposed by Anderson and Adolphs (2014), the speed increase caused by the electric shock may reflect a form of emotion. A series of candidate genetic analyses reveal that the response is not mediated by any single well-known chemo- or mechanosensory mechanisms. Instead, it requires voltage-gated calcium and potassium channel genes, which are required for electro-sensation in cartilaginous fishes (Bellono et al. 2017, 2018), suggesting an evolutionarily conserved mechanism for electro-sensation. Furthermore, we find that neuropeptide signaling regulates the duration of persistence. These results indicate that the response of *C. elegans* to electric shock can be a suitable paradigm to reveal genetic and physiological mechanisms of electro-sensation as well as persistent brain/behavioral states.

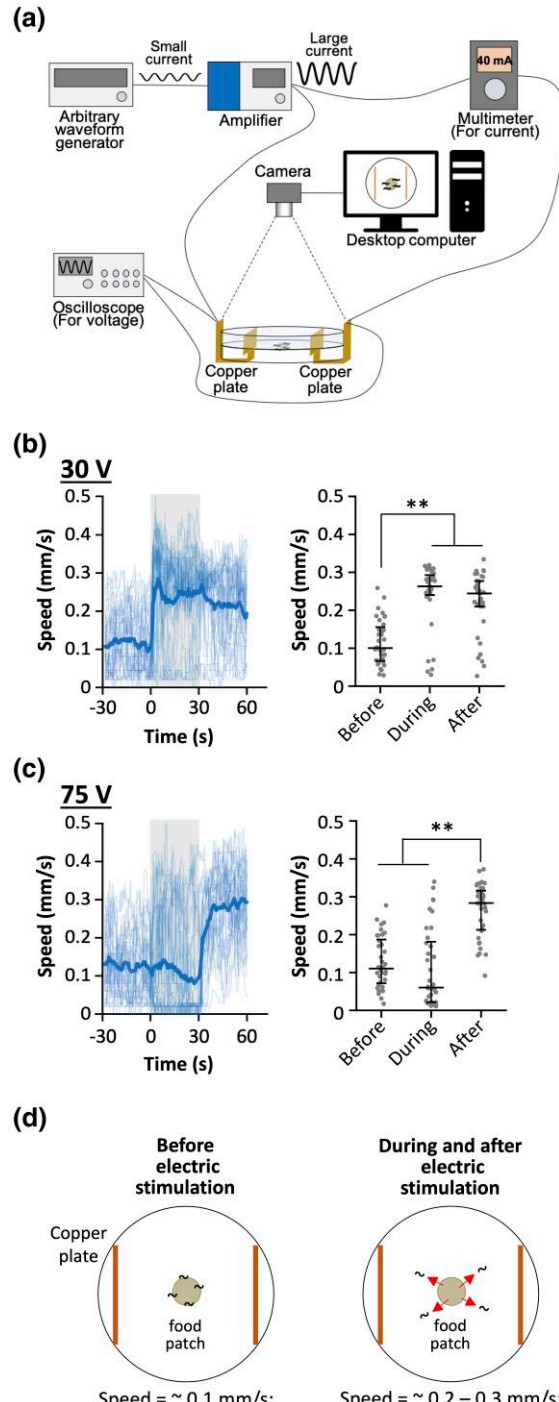

**Fig. 1.** Animals' speed is increased by AC stimulation. a) Experimental setup of electric shock experiment. This setup consists of an arbitrary waveform generator, amplifier, multimeter, oscilloscope, camera, and desktop computer. b) (Left) Speed–time graph with 30-V stimulation at 4 Hz. Thin and thick lines are for individual and average values, respectively. Gray indicates the duration of electric stimulation (0–30 seconds). (Right) Scatter plot showing average speed of individual animals before, during and after electric stimulation. Each period is 30 seconds. $n = 35$. c) Speed–time graph (left) and scatter plot (right) with 75-V stimulation at 4 Hz. $n = 36$. d) Cartoons of worm's response to the electric shock. (Left) Before electric stimulation, the animals stay on food patch and maintain their speed at around 0.1 mm/second. (Right) During electric stimulation delivery, the animals increase speed to around 0.2–0.3 mm/second and leave the food patch, which persists even after the stimulus is terminated. Statistical values were calculated using Kruskal–Wallis test with Bonferroni correction. **$P < 0.001$.

# Materials and methods

## *C. elegans* strains

*C. elegans* strains were maintained with standard procedures (Brenner 1974). In brief, for regular cultivation, animals were grown on standard 6-cm nematode growth medium (NGM) agar plates spread with *Escherichia coli* strain OP50 and incubated at 19.0–19.5°C. Strains used were the wild-type strain Bristol N2, mutant strains PR678 *tax-4(p678)*, CX4652 *osm-9(ky10);ocr-2(ak47)*, CB1033 *che-2(e1033)*, TU253 *mec-4(u253)*, ZB2551 *mec-10(tm1552)*, TQ296 *trp-4(sy695)*, MT1212 *egl-19(n582)*, DA995 *egl-19(ad995)*, JD21 *cca-1(ad1650)*, CB55 *unc-2(e55)*, VC854 *unc-2(gk366)*, NM1968 *slo-1(js379)*, BZ142 *slo-1(eg142)*, KDK11 *cat-2(tm2261)*, MT7988 *bas-1(ad446)*, GR1321 *tph-1(mg280)*, RB993 *tdc-1(ok914)*, VC671 *egl-3(ok979)*, MT1219 *egl-3(n589)*, KP3948 *lin-15B(n744); eri-1(mg366)*, and ZM5438. ZM5438 is the strain in which *Pmyo-3-egl-19* N-terminal cDNA was coinjected with a fosmid WRM0629dG07 in *egl-19(n582)* to produce the recombined *egl-19* minigene (Gao and Zhen 2011).

## *C. elegans* cultivation for electric shock behavioral assay

Before the behavioral assay, animals were cultivated as described previously (Kimura et al. 2010). In brief, 4 adult wild-type animals were placed onto NGM agar plates with OP50 and kept at 19.5°C for 7.5 hours before being removed. After removal, these plates were incubated at 19.0–19.5°C for 3 days until the assay day. On the assay day, about 100 synchronized young adult animals were grown on each plate. As some mutant animals had slower growth or laid fewer eggs than wild-type animals did, the incubation temperature and number of these mutant animals were adjusted and increased accordingly in order to obtain a developmental stage (i.e. young adult) and worm number comparable to the wild-type animals. All behavioral assays were carried out with young adult hermaphrodites.

## Experimental instruments for electric shock behavioral assay

The following electric instruments (Fig. 1a were utilized for the electric shock behavioral assay. A 50-MHz Arbitrary Waveform Generator (FGX-295, Texio Technology Corporation) was used to generate different types of electric waveforms over a wide range of frequencies. This waveform generator has an output limit of 10 V. Thus, an AC Power Supply (PCR500MA, Kikusui Electronics Corp.) was used to amplify the voltage supply. We also used a Digital Storage Oscilloscope (DCS-1054B, Texio Technology Corporation) in parallel to measure the voltage and observe the electric waveforms produced as well as a Digital Multimeter (PC720M, Sanwa Electric Instrument Co., Ltd.) to measure current. A USB camera (DMK72AUC02, The Imaging Source Co., Ltd.) with a lens (LM16JC5M2, Kowa) was used to record trajectories produced by the animals.

An increase in temperature with an electric stimulus was measured by a thermosensor AM-8000K equipped with XX-0203K-TS01 (Anritsu Meter Co., Ltd., Japan). The temperature in the agar at the 2 points close to the center was measured by putting the 2 sensor ends in the agar immediately before and after 1-minute or 5-second delivery of 30-V stimulus. The temperature changes from room temperature (~21.5°C) were $1.6 \pm 0.4$ (1 minute) or $0.0 \pm 0.0$ (5 seconds) (mean $\pm$ SD; $n = 5$).

## Electric shock behavioral assay with small OP50 bacterial food patch

Most of the behavioral assays were conducted on 9-cm NGM agar plates seeded with a small food patch unless otherwise indicated. For the food patch, the bacteria OP50 was grown in 100 mL of LB culture overnight at 37°C, spun down and resuspended in 10 volumes of NGM buffer, and 5 µL of the suspension was applied at the center of the plate to create a food patch $3 \times 10$ mm in size on the assay day. This process was used to minimize the thickness of the food patch as it prevents clear images of worms in the patch. Four animals per plate were placed in the food patch 1–3 hours before the assay to accustom the animals to the environment and to reduce their movement speed to the basal level. The assay plates were then inverted and placed onto a custom-made copper plate bridge, whose distance is 6 cm (Fig. 1). The images were acquired 2 frames per second, and electric shock was delivered with the conditions described in each figure. The assay was repeated 3–5 days per condition in general. Move-tr/2D software (Library Inc., Japan) was used to calculate the x–y coordinates of the animal centroids in each image frame, which were then analyzed using Excel (Microsoft) or R (The R Project) to calculate the animal's speed. The moving median for $\pm 1$ frame was calculated to remove noise for each animal and then ensemble averaged for each condition. Baseline speed was calculated from the average speed over 30 seconds before the stimulation, and $\Delta$speed was calculated by subtracting the baseline value from each animal's speed during or after the stimulus.

## Electric shock behavioral assay with full or strip-like OP50 bacterial food lawn

For the assays conducted with full food lawn, the region of the assay plates between the copper plates were fully seeded (about $5.5 \times 5.5$ cm$^2$) or seeded in a 3 stripe-shape (about $5.5 \times 1$ cm$^2$; Supplementary Fig. 4a) with OP50 and kept on the bench overnight until the assay began. A total of 60 µL or $20 \times 3$ µL of the OP50 suspension (see above) was used for the full and 3 stripe-shape food plates, respectively. Animals grown in regular cultivation plates were washed in 2 droplets of NGM buffer and then transferred to the center of the assay plate and left for 5 minutes. The rest of the procedures were the same as for assays conducted with small food patch.

To detect outward and inward movement on the food stripes (Supplementary Fig. 4), the food positions were first indicated on each image series by the experimenter and the animal's centroid across the boundary was automatically calculated by a custom-made program.

## Investigation of relationship among speed increase, current, and voltage

Three different types of NGM agar plates were prepared with varying salt concentrations and similar osmolarities: High-salt plates had 200-mM sodium chloride; low-salt plates had 10-mM sodium chloride and 380-mM sucrose; and control plates had 50-mM sodium chloride and 300-mM sucrose. The purpose of adding sucrose into the plates was to adjust and balance the osmolarity. The final total osmolarity for sodium chloride (Na$^+$ and Cl$^-$) and sucrose for all the plates was 400 mOsm. The rest of the procedures were the same as for assays conducted with small food patch.

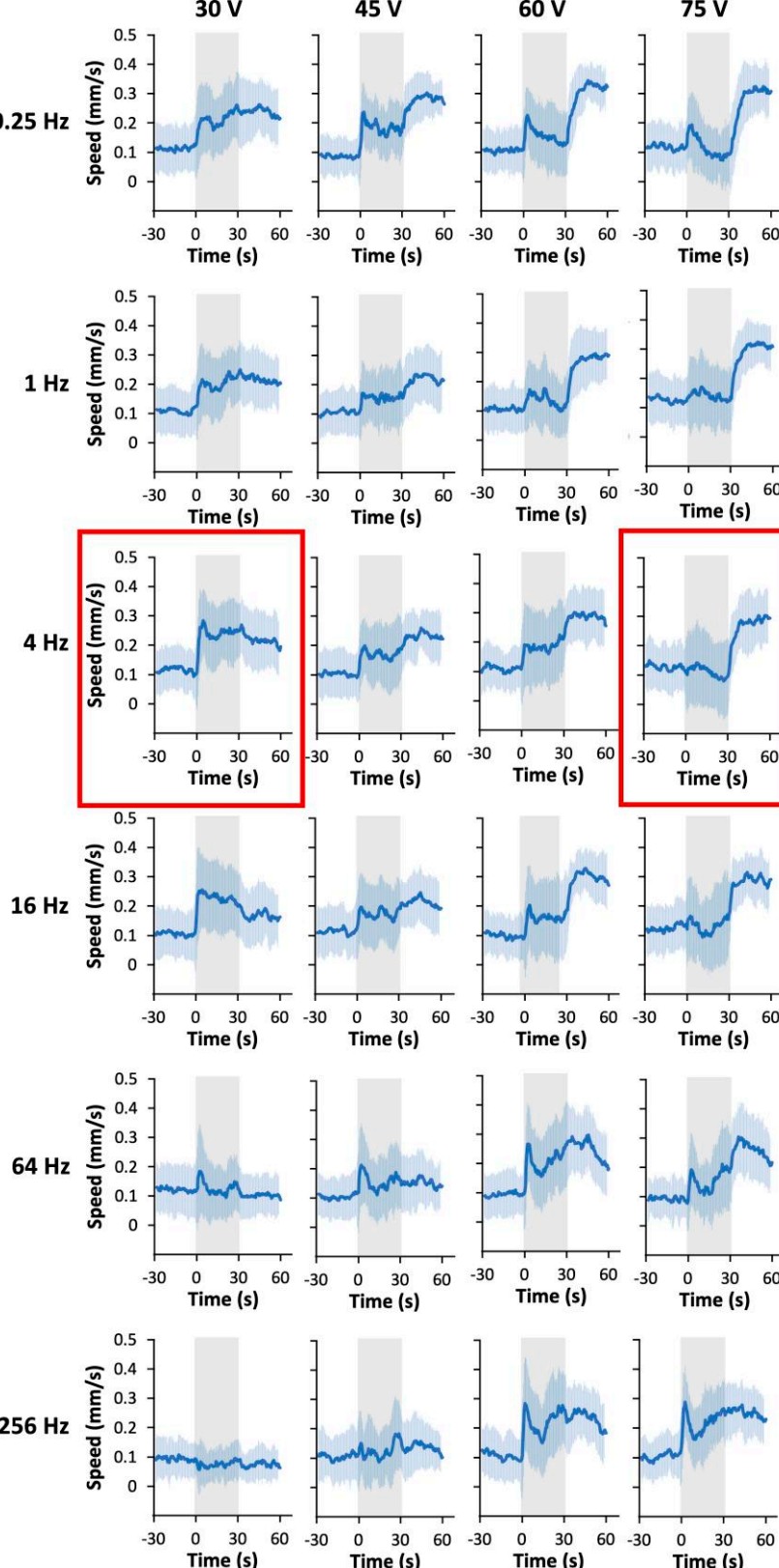

**Fig. 2.** Speed–time graphs with different voltage stimulation at different frequencies. Gray indicates the duration of electric stimulation (0–30 seconds). The thick line and the shaded region indicate the average ± SD. Thirty and 75 V at 4 Hz (red rectangles) were chosen for further analyses. Sample numbers were 33–37 per condition, and the details are described in the Supplementary Table 1.

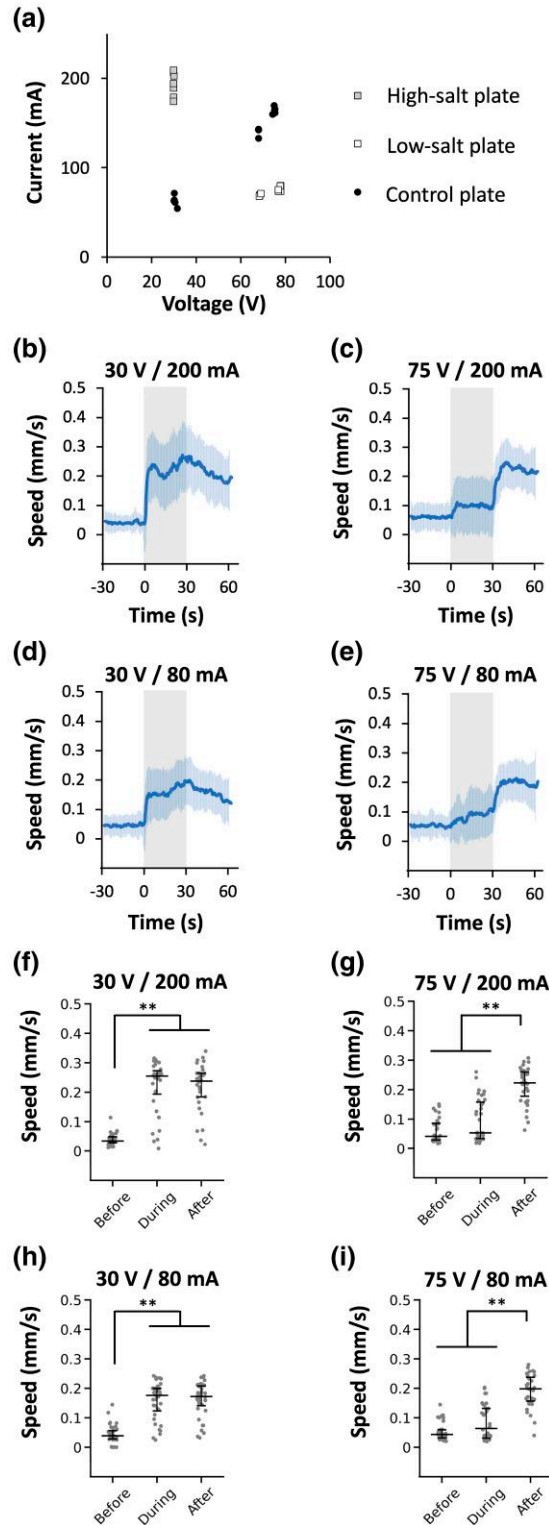

**Fig. 3.** Speed increase is dependent on voltage, not on current. a) Voltage–current graph with different salt concentrations (indicated by different symbols). Each dot represents the measured value on the day of the experiment. The final total osmolarity for sodium chloride (Na$^+$ and Cl$^-$) and sucrose for all the plates was 400 mOsm. b–e) Behavioral responses of animals assayed on high-salt plate with 30 V (b; $n = 32$), on control plate with 75 V (c; $n = 35$) or 30 V (d; $n = 36$), or on low-salt plate with 75 V (e; $n = 34$). Gray indicates the duration of electric stimulation (0–30 seconds). f–i) Scatter plot showing average speed of individual animals before, during and after electric stimulation, corresponding to panels b–e) respectively. Statistical values were calculated using Kruskal–Wallis test with Bonferroni correction. **$P < 0.001$.

## Suppression of EGL-19 activity

Feeding RNAi experiments were conducted according to previous studies with modifications (Timmons et al. 2001; Firnhaber and Hammarlund 2013). Briefly, 3 regions of 500–1,000 bp *egl-19* genomic fragments (the corresponding positions in Ch. IV as follows: "Fragment 1", 7,409,102–7,410,034; "Fragment 2", 7,410,308–7,410,875; and "Fragment 3", 7,411,365–7,412,174) were independently cloned in the NheI site of the L4440 vector, and the plasmid clones were named pMM89, pMM93, and pMM96, respectively. Each plasmid or empty L4440 was transformed into *E. coli* strain HT115. The transformed *E. coli* strains were cultured in 1 mL of LB medium with 100-µg/mL carbenicillin at 37°C overnight, and 75 µL or 300 µL (only for pMM96 containing HT115) of the culture was transferred to 3 mL of fresh LB medium and further cultured at 37°C until the $OD_{600}$ value reached 0.6–0.8. The grown culture was then spread on NGM plates containing 1-mM IPTG and 25- g/mL carbenicillin ("NGM–IPTG plate") and kept at room temperature for 1 day. Three L4 wild-type or mutant animals were placed on the plates and incubated at 21°C for 3 days. After 3 days, 10–14 animals ranging from L3 to L4 were moved to a new NGM-IPTG plate and incubated at 21°C overnight. Several young adult animals were transferred from the plates to the small food patch plate for subsequent electric shock behavioral assays. The "Fragment 3" *egl-19* plasmid clone was chosen because it had the strongest suppression effect on the electric shock-evoked speed increase. For all feeding RNAi experiments, control animals were fed with *E. coli* strain HT115 transformed with the empty L4440 vector.

For pan-neuronal RNAi by the expression of double-stranded RNA (Esposito et al. 2007), the sense or antisense "Fragment 3" of *egl-19* was fused with the *rab-3* promoter (Stefanakis et al. 2015) and *unc-54* 3′UTR using the PCR fusion method (Hobert 2002). The sense and antisense PCR fusion products (10 ng/µL each), IR101 (*rps-0p*::HygR::mCherry, 2 ng/µL), and sonicated OP50 genome (78 ng/µL) were coinjected (Mello et al. 1991) into wild-type animals to obtain the transgenic strains KDK55167 and KDK55221. IR101 was used for the selection of transgenic animals with hygromycin B (see below) (Radman et al. 2013). Injection of higher concentrations of the PCR fusion products of *egl-19* Fragment 3 did not generate transgenic animals for unknown reasons. Control transgenic lines (KDK55038 and KDK55054) without the *egl-19* plasmids were also obtained from the injection. For the behavioral analysis, 20 µL of 40-mg/µL hygromycin B (FUJIFILM Wako Chemicals Corp.) was added to an OP50-seeded NGM plate 1 day before egg-laying to obtain only animals with the transgene. The electric shock behavioral assays were conducted as described earlier.

## Data analysis and statistics

All the statistical analyses were performed in R (The R Project). Generally, data of 20–50 animals in total from 9 plates from 3 days of experiments for each condition were pooled and analyzed together. We chose this sample number based on a large scale behavioral analysis of *C. elegans* (Yemini et al. 2013). Data are presented as means ± SD unless otherwise specified. Experimental conditions, such as the electric stimulation or different strains were randomized on a daily basis.

## Results

### *C. elegans'* speed is increased by AC stimulation

Initially, we started this project by studying *C. elegans* behavioral responses to AC stimuli. The animals are known to respond to

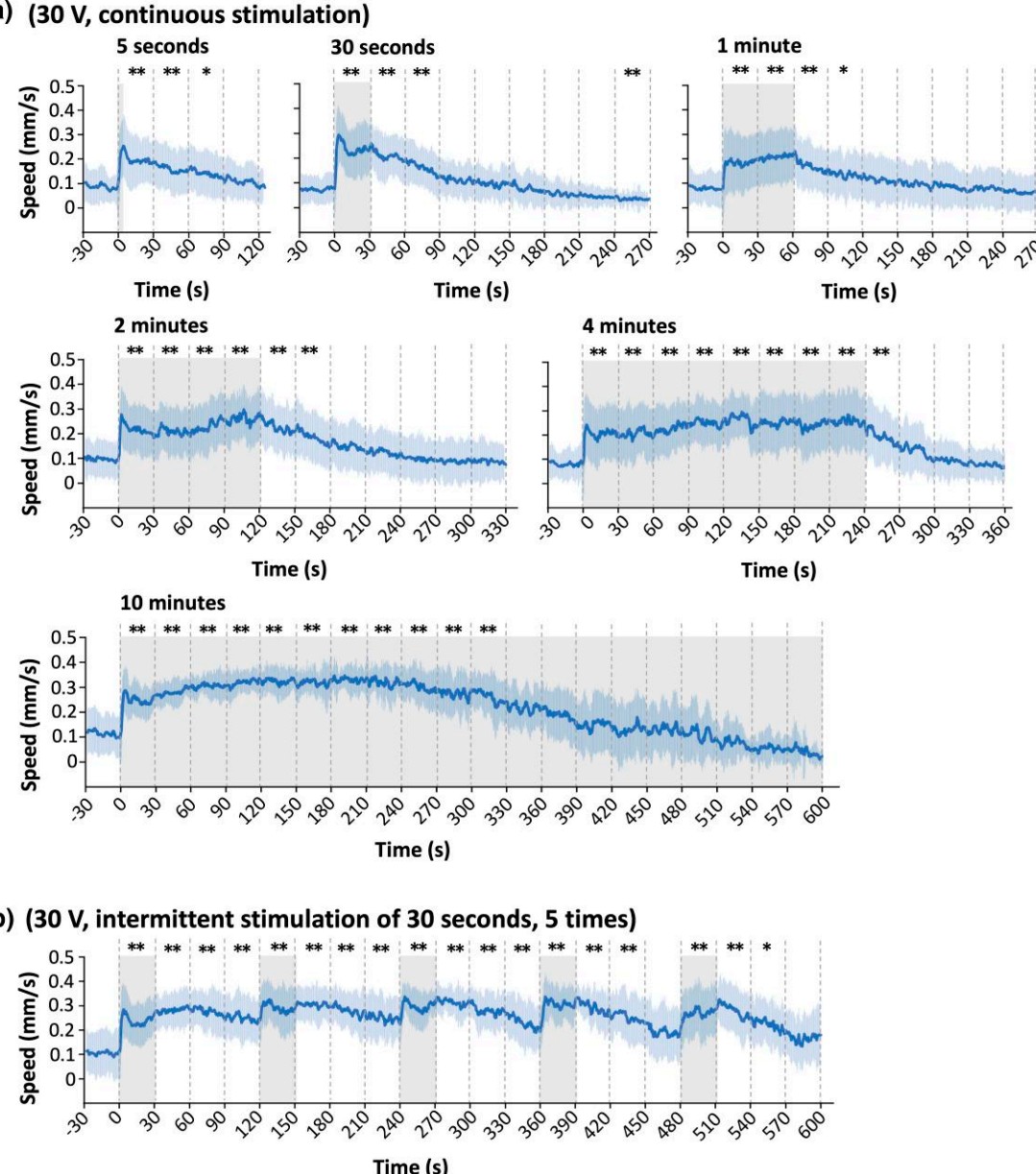

**Fig. 4.** Speed increase persisted for minutes even after the stimulation. a) Speed–time graphs of ON response with 30-V stimulation of different time periods, ranging from 5 seconds to 10 minutes. b) Speed–time graph for intermittent electric stimulation of 30 seconds, 5 times with 90-second intervals. c) Speed–time graphs of OFF response with 75-V stimulation of different time periods, ranging from 5 seconds to 1 minute. d and e) Speed–time graphs for electric stimulation of 30 V for 4 minutes d) or 75 V for 30 seconds e) with animals placed on full food lawn. Shaded regions around the lines represent standard deviation. Statistical values were calculated using Kruskal–Wallis test with Bonferroni correction for the differences from the average speed before the stimulation. *P < 0.01,** P < 0.001. Sample numbers were 32–46 per condition, and the details are described in Supplementary Table 1.

direct current (DC), migrating along the electric field from the positive end to the negative end (Sukul and Croll 1978), and a few classes of chemosensory neurons (ASH, ASJ, and AWC) were found to be required for their ability to align themselves according to the DC field (Gabel et al. 2007; Chrisman et al. 2016). However, the animal's migratory response to AC stimulus has not been reported yet. In our original setup (Fig. 1a), several adult wild-type animals were placed onto 9-cm agar plates seeded with a small bacterial food patch and subjected to AC stimulation. The complete trajectories produced by the animals were video-recorded, and their speed was calculated.

We first studied the response to AC stimulation covering a range between 15 and 105 V at 60 Hz (the commercial power

frequency in Japan), and found that the animals increased their average speed during electric stimulation by varying amounts (Supplementary Fig. 1). We then conducted a series of systematic analyses with different voltages and frequencies at 30–75 V and 0.25–256 Hz, and noticed that an interesting characteristic of this behavioral phenotype is most apparent when using 4-Hz stimuli: When animals were stimulated with 30 V, their average speed of movement suddenly increased more than 2-fold, and this persisted during and after the electric admission. We named this behavior the "ON response" (Fig. 1b and d). During this running behavior, the animals engage in rapid body bends as well as rapid head movements (Supplementary Videos 1 and 2). In the ON response, we did not detect a statistical bias in direction

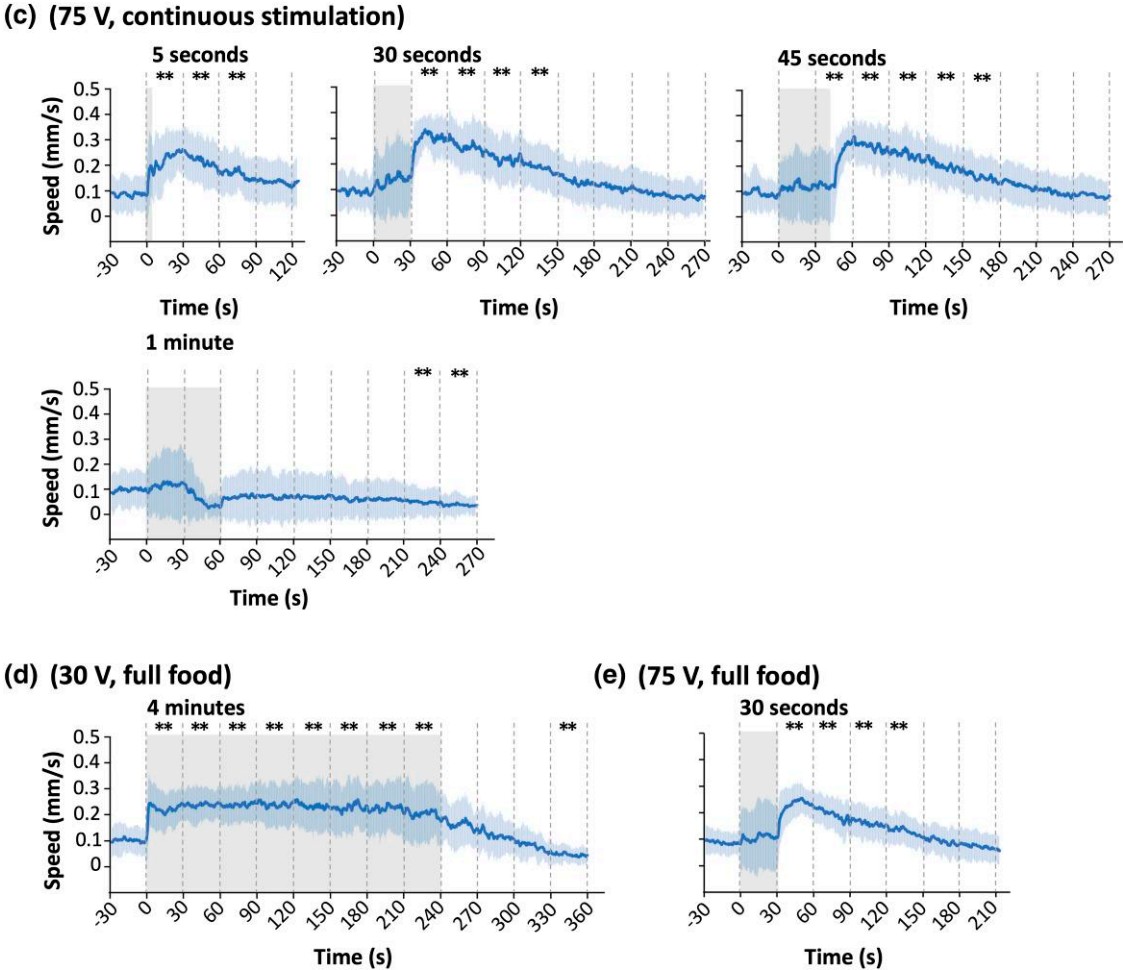

**Fig. 4.** Continued

(Supplementary Fig. 2). Unexpectedly, when a stronger electric stimulus of 75 V was applied, it caused a significant increase in average speed not during but immediately after the stimulus, which we named the "OFF response" (Fig. 1c). A fraction of the animals responded during the stimulus in the OFF response condition, while in the majority of the animals, the speed was suppressed during the stimulus and then increased immediately after its removal (Supplementary Fig. 3, Supplementary Videos 3 and 4); this behavioral difference may stem from variation in the threshold required to elicit the response. With other frequencies, ON and OFF responses were also observed but were less clear compared to those with 4 Hz (Fig. 2). The range of voltage per length (30–75 V/6 cm = 5–12.5 V/cm) is similar to the range previously shown to elicit responses to DC (3–12 V/cm) (Gabel et al. 2007), suggesting that these electric stimuli are physiologically meaningful for the animals.

We then analyzed whether this response depends on voltage or current by manipulating the salt concentration in the assay plate (Fig. 3): When 30 V is applied to the high-salt plate, the current should be similar to the current produced when 75 V is applied to a plate with our standard (control) salt concentration. Conversely, when 75 V is applied to the low-salt plate, the current should be similar to the current produced when 30 V is applied to the control plate (Fig. 3a). As shown in Fig. 3, 30- and 75-V stimuli caused ON and OFF responses, respectively, regardless of the current value, indicating that the behavioral response depends on voltage.

## Speed increase lasts for several minutes

Next, we examined how long the increased speed persists during and after the stimulus. When 30 V was applied for 0.5–2 minutes, significant speed increases were maintained during the stimulus, lasted for ~1 minute after the stimulus, then went back to the baseline level (Fig. 4a). Interestingly, when 30 V was applied for only 5 seconds, the speed increase still lasted for 1.5 minutes. When 4-minute stimulus was applied, the increase was maintained during the stimulus but went back to the baseline level 30 seconds after the stimulus. During 10-minute stimulation, the significant speed increase was observed only for 5.5 minutes. Thus, we concluded that the ON response caused by 30-V stimulation persists ~5 minutes at most.

This result suggested that the speed increase may decline after several minutes because of fatigue in motor systems. However, animals stimulated intermittently 5 times for 30 seconds per stimulation maintained a speed increase for a much longer time than those under the continuous stimulus (Fig. 4b vs "10 minutes" in 4a). This result supports the idea that the decrease in speed during the long ON stimulation period is not caused by fatigue in the motor system, but possibly by sensory adaptation, which is widely known to adjust the animal's sensory response to new environments (Wark et al. 2007).

We then tested the persistence of speed increase in the OFF response with 75 V. Five- and 30-second stimuli caused similar or

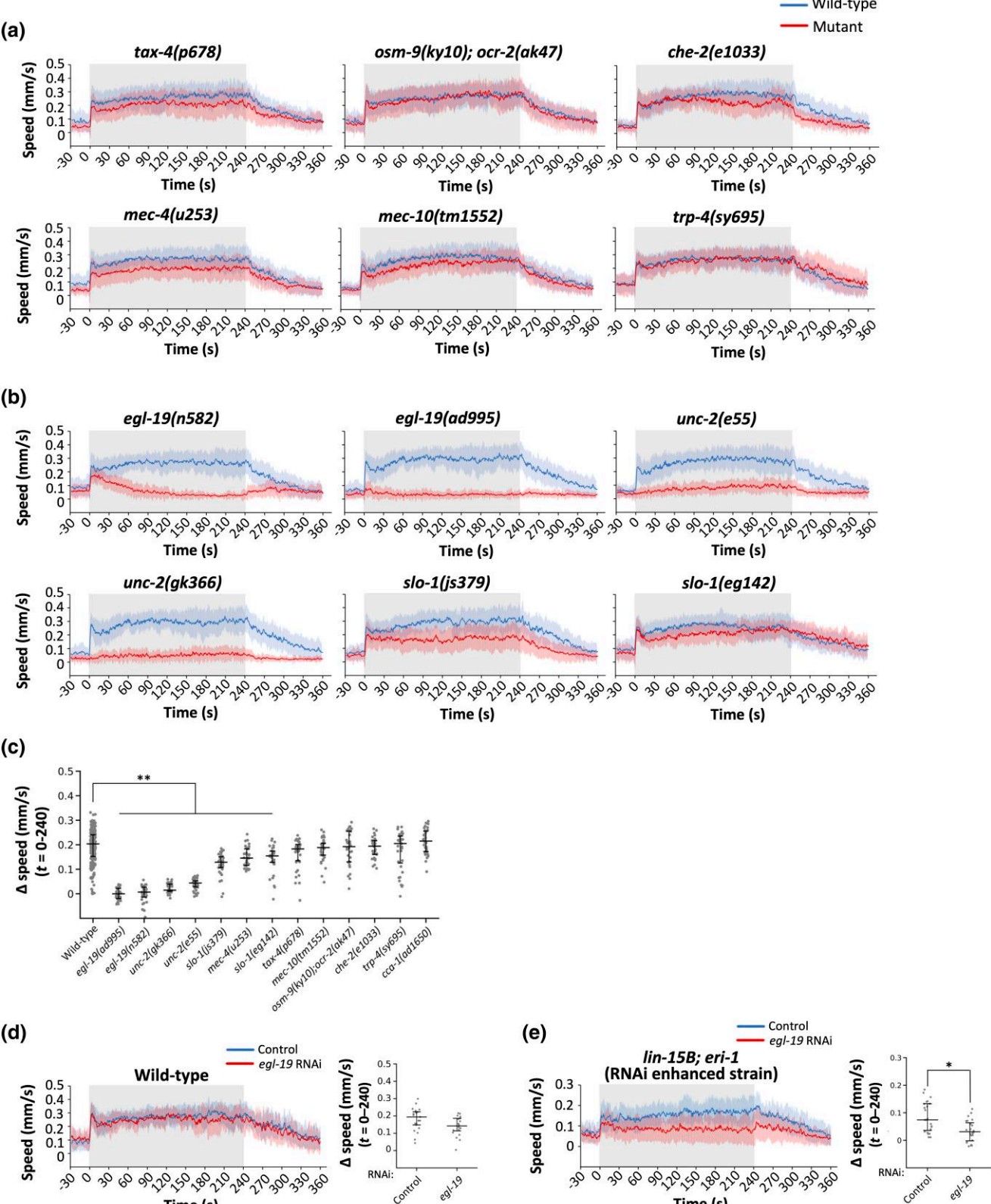

**Fig. 5.** Genetic analysis of ON response. a and b) Speed–time graphs of ON response with 30-V stimulation of 4 minutes on mutants of sensory signaling a) and of voltage-gated channels b). c) Scatter plot showing Δspeed of individual animals during 4 minutes of the stimulation (i.e. *t* = 0–240 seconds). In a series of daily experiments, wild-type animals and 3 to 5 mutant strains were analyzed in parallel. All the wild-type data are combined, and the mutant strains are arranged in ascending order of median values in c). d and e) Speed–time graphs (left) and scatter plot showing Δspeed (right) of ON response with 30-V stimulation of 4 minutes of wild-type d) or *lin-15B; eri-1* e) animals with (red or without (blue) feeding *egl-19* RNAi. f) Speed–time graphs (left) and scatter plot showing Δspeed (right) of ON response with 30-V stimulation of 4 minutes of 2 independent transgenic strains with (red) or without (blue) dsRNA of *egl-19* expressed under pan-neuronal promoter. g) Speed–time graphs (left) and scatter plot showing Δspeed (right) of ON response with 30-V stimulation of 4 minutes of wild-type (blue) or *egl-19(n582)* (green), or *egl-19(n582)* animals expressing the *egl-19* minigene only in muscles (red). Statistical values were calculated using Wilcoxon signed-rank test d and e) and Kruskal–Wallis test with Bonferroni correction c, f, and g). *P < 0.01, **P < 0.001. Sample numbers were 20–36 per mutant strain, and the details are described in the Supplementary Table 1.

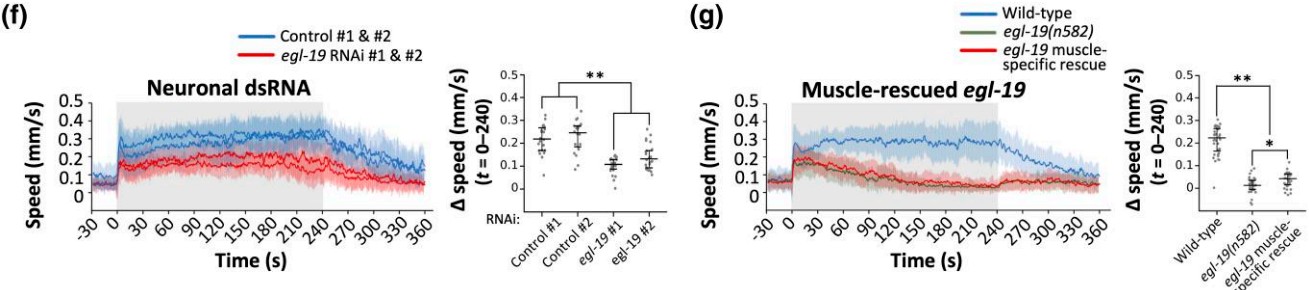

**Fig. 5.** Continued

longer persistent responses after the stimulus than 30 V did (Fig. 4c). Remarkably, 45-second stimulus caused >2 minutes persistent response, which is the longest among the responses to 30- and 75-V stimuli after the stimulus. When animals were stimulated for 1 minute, no ON or OFF responses were observed. The fact that the larger stimulus (75 V) caused longer persistent responses than the smaller one (30 V) suggests that the response to electric shock is "scalable" (i.e. different strength of stimulus causes different strength of behavioral response), one of the critical "emotion primitives" together with persistence (Anderson and Adolphs 2014).

We next tested the effect of food presence on the speed increase. *C. elegans* move slowly on the bacterial food lawn and faster out of the lawn (Sawin et al. 2000). As we used a small food lawn to localize the animal's initial positions to the center of the plate (Fig. 1a and d), it was possible that the electric stimulus caused the animals to move away from the food lawn, which then caused increased speed due to the absence of food. If this is the case, the animal's speed would be considerably lower with the electric stimulus when the plates were fully covered with a bacterial lawn. To test this hypothesis, we compared the time course of speed changes on plates with a small patch of food lawn and with a full food lawn. As shown in Fig. 4d and e (compare Fig. 4a "4 minutes" and c "30 seconds", respectively), there was no substantial difference in the time course of speed change between the small food and the full food plates in ON as well as OFF responses, demonstrating that the speed increase is not caused by the food absence but by the electric stimulation itself.

To further confirm that result, we analyzed the animals' speed on a 3-stripe food pattern (Supplementary Fig. 4a). We did not observe a significant difference in speed when the animals moved into or out of the food area (Supplementary Fig. 4b, Supplementary Video 5). This result suggests that the electric stimulus may have negative valence that is more influential to the animal's behavior than the food signal, even though food is critical for their survival. It further suggests that animals prioritize moving away from a harmful condition, such as the electric shock, to protect themselves.

## Voltage-gated ion channel genes are required for the AC response

The molecules required for responses to electric signals have only been revealed in cartilaginous fishes: Bellono et al. (2017, 2018) reported that electrosensory cells in little skate and chain catshark use L-type voltage-gated calcium channels (VGCCs) and voltage-gated big-conductance potassium (BK) channels. To identify gene(s) required for the response to electric shock in *C. elegans*, we analyzed a series of mutant strains of candidate genes. Specifically, we tested mutants of genes involved in the animals'

chemo- and mechanosensation, and the homologs of genes involved in electroreception in the cartilaginous fishes.

*C. elegans*' chemo-sensation is largely mediated by the 12 pairs of amphid sensory neurons in the head, classified into the ones using TAX-2 and TAX-4 cyclic nucleotide-gated channel (CNGC) subunits or the others using OSM-9 and OCR-2 transient receptor potential (TRP) channel subunits for depolarization (Coburn and Bargmann 1996; Komatsu et al. 1996; Colbert et al. 1997; Tobin et al. 2002). In addition to loss-of-function mutants for the above-mentioned genes, we tested mutants for *che-2*, a gene required for the proper formation and function of the sensory cilia (Fujiwara et al. 1999). For mechanosensation, we analyzed loss- or reduction-of-function alleles of *mec-4*, *mec-10*, and *trp-4*. *mec-4* and *mec-10* genes encode DEG/ENaC proteins and are responsible for the response of touch receptor neurons (Goodman and Sengupta 2019), while *trp-4* encodes TRPN (NOMPC) for harsh touch response (Kang et al. 2010). These mutant strains exhibited wild-type-like responses (Figs. 5 and 6, panels a and c); some mutants (*osm-9;ocr-2*, *che-2*, *mec-4*, *mec-10*, and *trp-4*) exhibited statistical differences in either ON or OFF response, suggesting the partial involvement of these genes, although the defects in speed increase (i.e. Δspeed) were not as severe as the ones of VGCC mutants (see below). The noninvolvement of *tax-4* also indicates that the temperature increase caused by the electric stimulus is not responsible for the speed increase; consistently, we did not observe any detectable temperature increase of the agar immediately after 30-V stimulation for 5 seconds (see Materials and methods for details).

We then tested *egl-19*, the ortholog of the L-type VGCC alpha subunit (Lee et al. 1997), which functions in the sensory organ for environmental electric signals for cartilaginous fishes (Bellono et al. 2017, 2018). We found that 2 reduction-of-function alleles of *egl-19* mutants exhibited strong defects in ON and OFF responses (Figs. 5 and 6, panels b and c).

Because *egl-19* is expressed in many neurons as well as muscles, we performed a series of experiments to clarify whether *egl-19* functions in neurons or in muscles. We first conducted feeding RNAi with wild-type animals and *lin-15B*; *eri-1*, a strain with enhanced RNAi, to overcome the problem of *C. elegans*' neurons resistance to feeding RNAi (Kamath et al. 2000; Timmons et al. 2001; Wang et al. 2005). With *egl-19* feeding RNAi, no effect was observed in wild-type animals. In contrast, *egl-19* feeding RNAi in *lin-15B*; *eri-1* caused significant suppression of the speed increase in ON response (Fig. 5d and e). Because feeding RNAi in wild-type animals should be effective in non-neuronal tissues, including muscles, these results suggest that *egl-19* functions not in muscles but in neurons. To confirm this finding, we conducted RNAi using the expression of double-stranded *egl-19* RNA under a pan-neuronal promoter, which causes neuron-specific RNAi (Esposito et al.

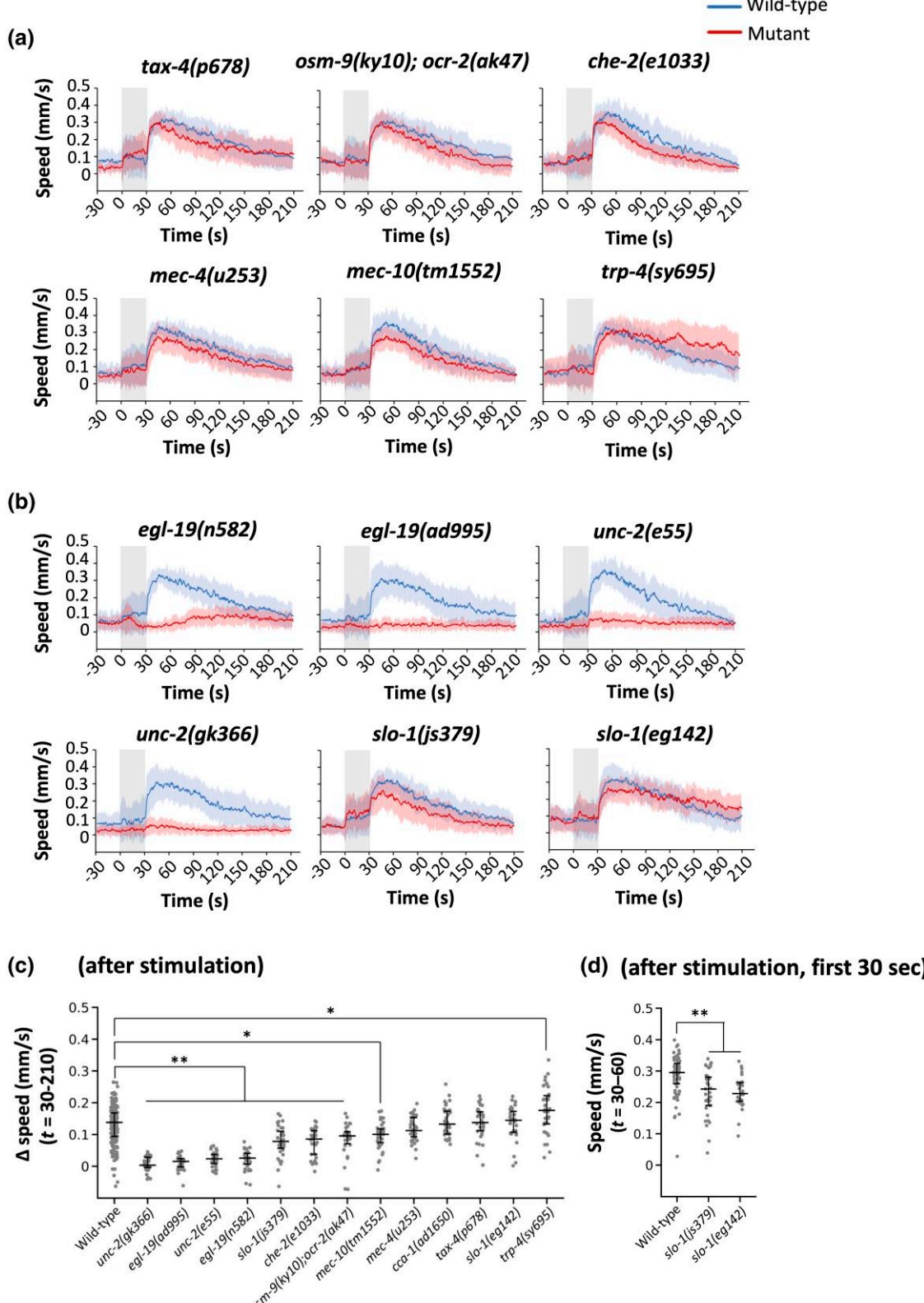

**Fig. 6.** Genetic analysis of OFF response. a and b) Speed–time graph of OFF response with 75-V stimulation of 30 seconds on mutants of sensory signaling a) and of voltage-dependent channels b). c) Scatter plot showing Δspeed of individual animals during 3 minutes after the stimulation (i.e. *t* = 30–210 seconds). In a set of daily experiments, wild-type and 3 to 5 mutant strains were analyzed in parallel. All the wild-type data are combined, and the mutant strains are arranged in ascending order of median values in c). d) Scatter plot showing the average speed of individual wild-type and *slo-1* mutant animals during 30 seconds after the stimulation (i.e. *t* = 30–60 seconds). Statistical values were calculated using Kruskal–Wallis test with Bonferroni correction. *P < 0.01, **P < 0.001. Sample numbers were 30–36 per mutant strain, and the details are described in the Supplementary Table 1.

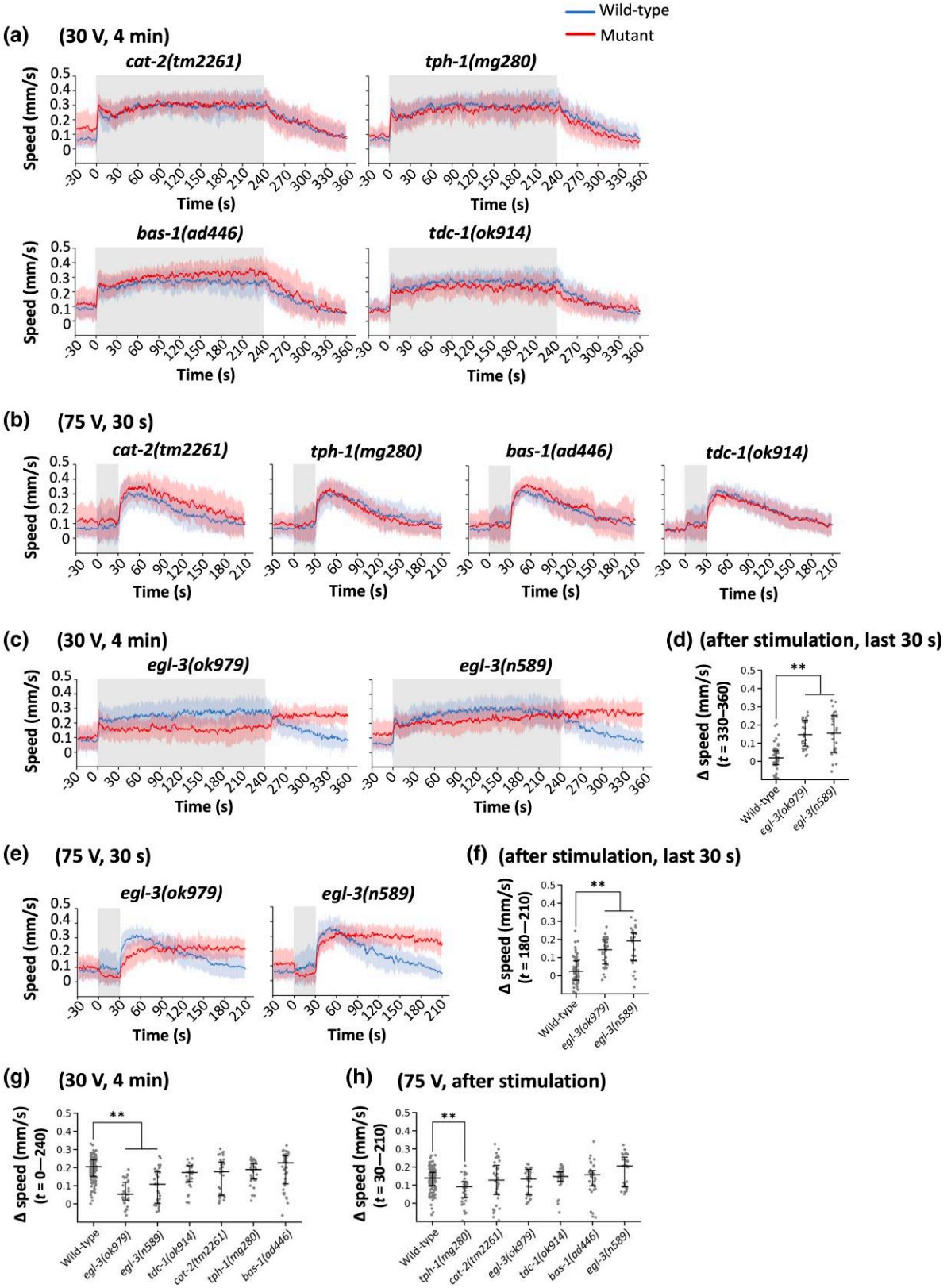

**Fig. 7.** Neuropeptides, but not other neuromodulators, are involved in the regulation of response persistence. a and b) Speed–time graphs of ON response with 30-V stimulation of 4 minutes a) or OFF response with 75-V stimulation of 30 seconds b) on mutants of biogenic amine biosynthesis. c and d) Speed–time graphs c) and scatter plot showing Δspeed d) of ON response with 30-V stimulation of 4 minutes on 2 alleles of *egl-3* mutants. e and f) Speed–time graphs e) and scatter plot showing Δspeed f) of OFF response with 75-V stimulation of 30 seconds on 2 alleles of *egl-3* mutants. The time used for scatter plot was t = 330–360 seconds d) or 180–210 seconds f). g and h) Scatter plot showing Δspeed of individual animals during 4 minutes of g) or 3 minutes after h) the stimulation (i.e. t = 0–240 seconds or t = 30–210 seconds, respectively). Statistical values were calculated using Kruskal–Wallis test with Bonferroni correction. **P < 0.001. Sample numbers were 29–36 per mutant strain, and the details are described in the Supplementary Table 1.

2007). As shown in Fig. 5f, this caused significant suppression of the ON response. Furthermore, expressing an *egl-19* minigene only in the muscles of *egl-19* mutants (Gao and Zhen 2011) did not rescue the phenotype (Fig. 5g). All of these results are consistent with the idea that *egl-19* functions in neurons but not in muscles. One allele of *egl-19* mutants exhibited movement speed comparable to wild-type animals before stimulation (Supplementary Fig. 5), suggesting that the defect in the speed increase is not caused by a problem in the basal locomotory system.

The involvement of *egl-19* in the response to electric shock further motivated us to test 2 other types of VGCCs, namely, N-type (UNC-2) and T-type (CCA-1) VGCCs (Schafer and Kenyon 1995; Steger et al. 2005), although only L-type VGCC had been found to be involved in electrical responses in the cartilaginous fishes. Unexpectedly, mutants for 2 alleles of *unc-2* were defective in both ON and OFF responses, while *cca-1* mutants behaved similar to the wild-type controls (Fig. 5 and 6, panels b and c, and Supplementary Fig. 6).

We then investigated the involvement of the BK channel, a voltage-gated potassium channel, also known to be involved in electro-sensation in cartilaginous fish (Bellono et al. 2017, 2018). Interestingly, 2 alleles of *slo-1*, the sole ortholog of BK channels in *C. elegans* (Wang et al. 2001; Davies et al. 2003), also exhibited statistical differences in the ON as well as the OFF response, at least in some aspects (Fig. 5b and c for ON response and Fig. 6b–d for OFF response). The possible involvement of BK channels in addition to the L-type VGCC in *C. elegans*' electrical response suggest that the molecular mechanisms of electro-sensation may be evolutionarily conserved, although a novel component (N-type VGCC) is also involved.

## Neuropeptide signaling down-regulates the duration of persistent response

Lastly, we attempted to identify genes required for behavioral persistency, and considered the genes involved in the biosynthesis of neuromodulators as candidates. We tested *cat-2* (dopamine), *tph-1* (serotonin), *bas-1* (dopamine and serotonin), and *tdc-1* (tyramine and octopamine) mutant animals (Loer and Kenyon 1993; Lints and Emmons 1999; Sze et al. 2000; Alkema et al. 2005), and most of these mutants exhibited wild-type-like responses, indicating that these neuromodulators are not involved (Fig. 7a, b, g, and h); although *tph-1* mutants exhibited a statistical difference in OFF response, its contribution does not appear substantial. Because dopamine and serotonin signaling are known to be required for the feeding status-dependent modulation of migratory speed, these results are also consistent with the fact that feeding status is not the causal reason for the speed increase (Fig. 4d and e, and Supplementary Fig. 4).

We then further tested the involvement of neuropeptides by using loss- or reduction-of-function mutations of *egl-3*, a gene required for maturation of pro-neuropeptides (Kass et al. 2001). Unexpectedly, mutations in both alleles of *egl-3*, *n589*, and *ok979*, caused weaker 30-V ON response and, moreover, much longer persistence of the speed increase after the electric shock in ON and OFF responses (Fig. 7c–h), indicating that the persistent activity in the neural circuit for speed increase is down-regulated by neuropeptide signaling in the wild-type animals.

## Discussion

In the present study, we revealed that *C. elegans* exhibits a persistent speed increase in response to AC stimuli. This behavioral response appears characterized by persistence, scalability, and valence, suggesting that it may reflect an emotional state of *C. elegans*, which has never been reported. In addition, genetic analysis revealed that genes involved in electro-sensation in cartilaginous fishes and a neuropeptide biosynthesis gene are required for the response, demonstrating that this behavior is an ideal paradigm for genetic dissection of both electro-sensation and persistent behavioral states.

## Response to electric stimulus and its mechanisms in *C. elegans* and other animal species

In neuroscience research, electricity is used as an unconditioned stimulus with negative valence to cause associative learning in rodents and flies (Rescorla 1968; Quinn et al. 1974). In nature, however, multiple animal species are known to respond to electricity for survival purposes, such as communication, navigation and/or prey detection (Pettigrew 1999; Crampton 2019). For example, weakly electric African fish (*Gnathonemus petersii*) utilize their epidermal electroreceptors to receive self-produced electric signals, allowing the fish to identify, locate, and examine nearby objects (von der Emde et al. 2008). In addition, platypuses (*Ornithorhynchus anatinus*) detect electric signals via their duck-like bills to locate and avoid objects when navigating in the water (Scheich et al. 1986). Blind cave salamanders (*Proteus anguinus*) perceive a moving back-and-forth direct-current field and its polarity via ampullary organs to survive and navigate in their environment, which is in complete darkness as their eyes are undeveloped (Istenič and Bulog 1984; Roth and Schlegel 1988). In invertebrates, bumblebees (*Bombus terrestris*) sense environmental electric fields via sensory hairs to make foraging decisions (Clarke et al. 2013; Sutton et al. 2016). In a recent study, *C. elegans* is also shown to exhibit phoretic attachment to other insects by nictating and transferring across DC electric fields (Chiba et al. 2023). Such diverse usage of electric signals, across a range of animal taxa, suggests that detecting and responding to electric signals is of broad importance, yet the underlying molecular mechanisms remain poorly understood.

In this study, we established an original experimental paradigm and found that *C. elegans* responds to an AC electric stimulus: The animals significantly increase their movement speed during and after the stimulus for minutes. Although the animals have also been reported to respond to and utilize DC (Gabel et al. 2007; Chrisman et al. 2016; Chiba et al. 2023), we consider that the responses to AC and DC are substantially different for the following reasons. (1) In the DC field, the animals moved at a certain angle (~4° per 1 V/cm), which was not observed in our AC stimulus (Supplementary Fig. 2). (2) Movement speed did not change with the DC stimulus (Gabel et al. 2007). (3) Another DC response of worms, described as "nictating-and-leaping", involves worms being moved passively by the electricity (Chiba et al. 2023), whereas the AC response we report is actively directed by the worm.

In addition, although 3 pairs of amphid sensory neurons play important roles in the DC response (Gabel et al. 2007; Chrisman et al. 2016), mutations in genes required for sensory signaling in amphid sensory neurons (*tax-4*, *osm-9*, *ocr-2*, and *che-2*) did not affect the AC response substantially in our study (Figs. 5 and 6), indicating that DC and AC responses utilize different sensory mechanisms. This result also rules out the possibility that the animals respond to increased agar temperature due to the AC stimulus, because *tax-4* is essential for temperature sensation (Komatsu et al. 1996) and because the plate temperature did not increase, at least after 30-V stimulation for 5 seconds (see *Materials and methods* for details). The genes required for mechanosensation (*mec-*

4, *mec-10*, and *trp-4*) do not seem to play a critical role in the AC response either. Still, it is possible that the AC stimulus is sensed by multiple types of sensory neurons redundantly.

We found that the VGCC and possibly the BK channel, the voltage-gated calcium and potassium channels for electrosensation in cartilaginous fishes, are involved in the AC response of *C. elegans*. The involvement of multiple types of voltage-gated channels, homologous across distantly related species, in the sensation of electricity suggests that this mechanism is evolutionarily conserved. It also suggests that EGL-19 and SLO-1 may function coordinately in a subset of neurons that sense electricity. Indeed, our data indicate that *egl-19* functions in neurons instead of muscles in this behavioral response (Fig. 5d–g). Since *egl-19* and *slo-1* are widely expressed in most neurons (Lee et al. 1997; Wang et al. 2001; Davies et al. 2003), it would be interesting to identify the neurons where these genes function to sense the electric signals. It should be noted that the N-type VGCC UNC-2 is also essential for the response to electric shock (Fig. 5b and 6b), suggesting that mechanisms of electric sensation may be more diverse among animal species.

## Electric stimulus causes persistent behavioral response

Persistent neural activity, a sustained neural activity caused by a short-term stimulus, plays critical roles in brain function, such as controlling motivation, arousal, and emotion as well as working memory and decision-making, although its detailed mechanisms have not been sufficiently elucidated (Berridge 2004; Major and Tank 2004; Curtis and Lee 2010; Anderson 2016). Persistent behavioral state is caused by persistent neural activity, suggesting that genetic analysis of persistent behavioral state can reveal molecular mechanism(s) of persistent neural activity that underlies brain functions.

We unexpectedly found that *C. elegans'* high speed response persists after electric shock. In *C. elegans*, 2 other types of persistent behavioral responses related to speed change have been reported. The first is that the animal's movement speed is elevated at high $O_2$ concentration in *npr-1*(*lf*) and in the Hawaiian wild isolate CB4856, which has the same amino acid variation in *npr-1* (Cheung et al. 2005). In this behavioral response, (1) the elevated speed returns rapidly to the basal speed when the high $O_2$ is terminated, (2) the animals still recognize and aggregate at the edge of a food lawn, and (3) a mutation in the *tax-4* CNGC homolog for sensory depolarization abolishes the response (Coates and de Bono 2002). Another type of persistent behavioral response is roaming (Fujiwara et al. 2002; Flavell et al. 2020). Roaming is a behavioral state with high movement speed, although it is only exhibited when the animals are on food and requires serotonin signaling. Because the behavioral response to electric shock persists more than 2 minutes after 30–45 seconds stimulus with 75 V and more than 1.5 minutes after only 5-second stimulus, is not affected by food stimulus, and does not require CNGC activity or serotonin signaling, electric shock response is likely different from the above-mentioned two behavioral responses, and its analysis may provide a unique opportunity for genetic dissection of a persistent behavioral state and neural activity.

Interestingly, we revealed that the persistent aspect of the behavioral response is down-regulated by *egl-3*, a gene required for maturation of pro-neuropeptides (Kass et al. 2001), which affects biosynthesis of FMRFamide-like peptide (FLP) and neuropeptide-like protein (NLP), but not insulin-like peptides (ILP) (Husson et al. 2007). Because the requirement of neuropeptide signaling

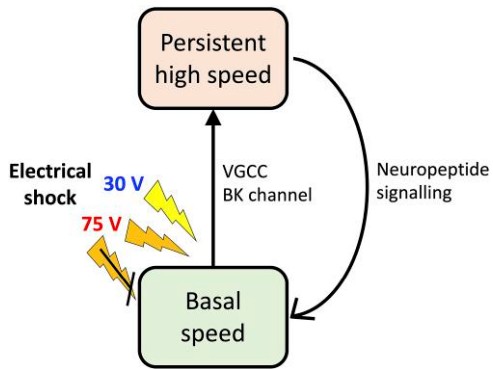

**Fig. 8.** Model for mechanism of speed increase caused by electric shock.

is reminiscent of neuropeptide regulation of fear in mammals including humans (Bowers et al. 2012; Comeras et al. 2019; van den Burg and Stoop 2019), the fear-like brain state may be regulated by evolutionarily conserved molecular mechanisms.

Electric shock is widely utilized as an unconditional stimulus in fear conditioning paradigms, especially in rodents, where less than 1 mA of current for a few seconds is generally used (Korte et al. 1999; Toth et al. 2012). Thus, the conditions used in this study (80–200 mA in current; Fig. 3) may appear artificial. However, we consider that the responses of *C. elegans* to these stimuli reflect physiologically meaningful biological mechanisms for the following reasons: (1) The range of voltage per length (30–75 V/6 cm = 5–12.5 V/cm) is similar to the one used to study the animal's DC response (3–12 V/cm) (Gabel et al. 2007). (2) The electric current flowing inside the worm's body could be weak because it depends on the resistance of its body and cuticle. (3) Only a 5-second stimulus causes a persistent response that lasts more than a minute, meaning that the electric shock itself is just a trigger and what we observe is a physiological response to that trigger. The speed increase behavior we observed may resemble fleeing, one of the most common responses caused by fear in higher animals and humans (Adolphs 2013; Mobbs and Kim 2015; Bliss-Moreau 2017).

## Response to the electric stimulus may reflect a form of emotion

Emotions are internal brain states triggered by certain types of environmental stimuli, which are associated with cognitive, behavioral, and physiological responses (Nettle and Bateson 2012; Anderson and Adolphs 2014; Perry and Baciadonna 2017; Abbott 2020). Recently, multiple species of invertebrates are considered to possess internal brain states that resemble what we consider to be emotions (Fossat et al. 2014; Gibson et al. 2015; Hamilton et al. 2016; Mohammad et al. 2016; Solvi et al. 2016; Bacqué-Cazenave et al. 2017). One of the most prominent characteristics of emotion across animal species is its persistence: For example, even a transient environmental stimulus can cause a persistent behavioral response, such as courtship, aggressive, and defensive behavior (Nettle and Bateson 2012; Anderson and Adolphs 2014; Perry and Baciadonna 2017; Paul and Mendl 2018; Abbott 2020).

Anderson and Adolphs proposed a new framework to study emotions across animal species, wherein hallmarks of an emotional state are persistence, scalability, valence, and generalization. In addition to persistence (Fig. 4a–c), we consider that the electric response has negative valence. This is because the animals ignore food during the electric shock response (Fig. 4d and

e, and Supplementary Fig. 4), despite the fact that food is one of the most influential signals for *C. elegans*, affecting many aspects of their behavior. For example, during the high speed state caused by high $O_2$, animals still recognize and stay at the edge of a food lawn (Coates and de Bono 2002; Cheung et al. 2005), suggesting that the electric shock signal has a strong negative valence that overrides the strong positive valence of food. The third point is the scalability—stronger stimulus causes stronger behavioral response. Compared to the 30-V stimulus, the 75-V stimulus results in a longer-lasting high speed response after the stimulus (Fig. 4a and c). The fourth point is generalization—the same emotional state can be triggered by different stimuli and, in turn, the emotional state triggered by one stimulus can then affect responses to other stimuli. The lack of response to food during and following our electric stimulus might support this point as well, as the emotional state induced by electricity influences the response to food, an entirely different stimulus. Taken together, these results may suggest that the animal's response to electric shock represents a form of emotion, possibly akin to fear.

In summary, we found that *C. elegans* persistently responds to electric shock, which is regulated by voltage-gated ion channels and neuropeptide signaling. Our findings suggest the following model (Fig. 8). When the animals sense 30- or 75-V AC stimulus at 4 Hz, the stimulus is sensed with the VGCC and BK channel and their internal state transits from basal speed state to persistent high speed state. The persistent high speed state eventually returns to the basal speed state, which requires neuropeptide signaling. By taking advantage of connectome information and the methods for imaging whole brain activity of identified neurons (White et al. 1986; Randi and Leifer 2020; Wen et al. 2021; Yemini et al. 2021), *C. elegans* may become one of the ideal models for revealing the dynamic information processing involved in the entire neural circuit related to emotion.

## Data availability

Strains and plasmids are available upon request. The authors affirm that all data necessary for confirming the conclusions of the article are present within the article, figures, and tables. The data of worms' coordinates used in all the figure panels are available in https://ssbd.riken.jp/repository/315/.

Supplemental material available at GENETICS online

## Acknowledgments

We thank Liting Chen for having provided the idea of "worm's emotion" for KDK, Yuki Tanimoto and Yuka Tsuda for the initial phase of the electric shock paradigm, Shinobu Aoyagi for setting up the system, Chentao Wen for developing the outward/inward scoring program, Jason Chin and Mei Zhen for plasmids and strains, Yukimasa Shibata and Kiyoji Nishwaki for teaching feeding RNAi, and Masahiro Tomioka, Hirofumi Kunitomo, Young-Jai You, Aki Takahashi, Atsuko Nakagawa, and the Kimura lab members for their valuable advice, comments, and technical assistance for the study. Nematode strains were provided by the Caenorhabditis Genetics Center (funded by the NIH Office of Research Infrastructure Programs P40 OD010440).

## Funding

This study was supported by Japan Society for the Promotion of Science (JP16H06545, 20H05700, 21H00448, 21H02533, 21H05299, 21K19274, and 22KK0100 to KDK), Grant-in-Aid for Research in Nagoya City University (48, 1912011, 1921102, and 2121101), the Joint Research by National Institutes of Natural Sciences (01112002, 22EXC206, 23EXC204), Toyoaki Scholarship Foundation, and RIKEN Center for Advanced Intelligence Project (to KDK). LFT was supported by Japanese Government (MEXT) Scholarship .

## Conflicts of interest

The author(s) declare no conflict of interest.

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

*Editor: A. Barrios*