## [Peer Review File · Genetics]

Electric shock causes a fleeing-like persistent behavioral response in the nematode *Caenorhabditis elegans*

Ling Fei Tee, Jared Young, Ryoga Suzuki, Keisuke Maruyama, Sota Kimura, Yuto Endo, and Koutarou Kimura

NOTE: The reviews and decision letters are unedited and appear as submitted by the reviewers.

In extremely rare instances and as determined by a Senior Editor or the EIC, portions of a review may be redacted. If a review is signed, the reviewer has agreed to no longer remain anonymous.

The review history appears in chronological order.

Review Timeline:

Submission Date:	2022-04-11
Editorial Decision:	2022-05-11
Resubmission Received:	2023-07-05
Editorial Decision:	2023-07-21
Revision Received:	2023-07-26
Accepted:	2023-07-27

May 11, 2022

GENETICS-2022-305222

Electric shock causes a fear-like persistent behavioral response in the nematode *Caenorhabditis elegans*

Dear Dr. Kimura:

Two experts in the field and myself have reviewed your manuscript. I am pleased to inform you that, with minor revisions, it is potentially suitable for publication in GENETICS. The reviewers have comments and concerns that need to be addressed in a revised manuscript. You can read their reviews at the end of this email.

It is most important that you address the following in your resubmission. The evidence for the behavioural response to AC being a primitive for fear is not strong enough yet. It is unclear whether the behavioural response has negative valence and whether it can be generalised (two of the characteristics extracted by Anderson and Adolphs). One way to address both of these is to test whether the AC treatment can negatively condition a stimulus (salt, benzaldehyde or butane would be good candidates) in a simple associative learning task. In addition, some knowledge of the neurons involved in sensing or executing the behavioural responses to AC would highly improve the manuscript. There are some suggestions about how to do this by the reviewers below.

When you resubmit, please let the editorial office know approximately how long you expect to need for revisions.

Upon resubmission, please include:

1. A clean version of your manuscript;
2. A marked version of your manuscript in which you highlight significant revisions carried out in response to the major points raised by the editor/reviewers (track changes is acceptable if preferred);
3. A detailed response to the editor's/reviewers' comments and to the concerns listed above. Please reference line numbers in this response to aid the editors.

Additionally, please ensure that your resubmission is formatted for GENETICS.

<https://academic.oup.com/genetics/pages/general-instructions>

Follow this link to submit the revised manuscript: Link Not Available

Sincerely,

Arantza Barrios
Associate Editor
GENETICS

Approved by:
Oliver Hobert
Senior Editor
GENETICS

Reviewer #1 (Comments for the Authors (Required)):

Tee et al provide a fun study that explores the genetic basis for how *C. elegans* responds to AC electrical stimuli. They tested a wide range of parameters in WT and many different mutant backgrounds. Their most salient results are that certain AC stimuli cause the worm to maintain its speed during stimulation but speed up afterward, whereas strong stimuli tend to cause the worm to speed up only during stimulation. These responses appear to require calcium (*egl-19* and *unc-2*) and potassium (*slo-1*) channels, which somewhat matches how other electrically sensitive animals sense electrical stimuli. The characteristic decline of response after stimuli appears to require an unspecified collection of neuropeptides because *egl-3* mutants exhibited an unusually long response. The study is well conducted and controlled. The statistics appear appropriate.

Major issues:

1. It would have been interesting to have determined which sensory neurons, if any, are required for these AC responses; however, they found that many sensory pathways appear to be dispensable for this response. This raises the possibility that multiple sensory pathways sense AC redundantly in parallel, and they missed this result because they didn't break the right

combination of sensory neurons. This needs to be stated in the Discussion. Alternatively, sensory neurons might not even be required to respond. Indeed, the calcium and potassium channels that are required for response are expressed throughout the nervous system. I wonder if they could use cell-selective expression of RNAi to knock-down *egl-19* in at least a large subset of sensory neurons to deduce whether sensory neurons are in fact required for this response? Or use RNAi by feeding in a worm strain that expresses *sid-1* in sensory neurons? This would be very interesting. Some useful public strains include: TU3403 (touch neurons), XE1375 (GABA), XE1474 (dopamine), XE1581 (ACh), and XE1582 (glutamatergic).

2. I cannot tell from your experiments if the heightened speed during or after shock reflects "fear" or other types of emotion. Do any of the AC stimuli serve as effective unconditioned cues for associative learning? This can be easily tested. Without testing this, I find it hard to know if the worms are "excited", "happy", or "fearful" when maintaining their speed during AC or moving fast afterwards. Why not test whether worms form a negative association with innately attractive NaCl in the agar substrate with a chemotaxis assay after training. Then you might have clear results. There are other useful ways to test for negative association too. Alternatively, I would be happy if they just toned down the idea of fear without more experimental evidence.

Minor issues:

Line 67. "express" probably should be rewritten as "reflect"

Line 68. ...the response if not mediated by any single well-known... (you need to say this because they may be mediated by more than one in redundantly in parallel)

Line 73. ...indicate that the response of *C. elegans* to electric shock...

Line 83. Please add info here and elsewhere in Results and Discussion from Chrisman et al., 2016. They got distinct results from Gabel et al. 2007

Line 100. Was there any mutant that speeds up *during* this stimuli?

Lines 109-113. This is confusing as written. I'd recommend taking out this out here and elaborate more carefully in the Discussion.

Figure 2A and B. The thick lines are supposed to reflect individual measurements and not SEM. Then why does absolute speed measurements appear to drop below zero? This doesn't make sense.

Figure 2C. I'm confused why you needed to have worms start on the food. Why not have the worms start anywhere. The presence of the food seems to complicate interpretation of results because you do not clarify in most cases when worms are on or off food.

Figure 4A. I'd recommend adding sublabels to help explain the six results you have within this single panel. For instance, the bottom one where you stimulate for 600 seconds is interesting. You speculate that the worms may be experiencing exhaustion. To determine this, you might try using laser blue light to stimulate the worms. This may dishabituate the worms and cause them to speed up again, or not.

Figure 5. You could try backing up your *egl-19* results with L-type calcium channel blocker that effects *C. elegans*. This would test if you get the same result even if you transiently block function of the channel rather than lack EGL-19 channel the entire life of the animal.

Figure 5 E and F. You are measuring different time windows in these two panels. To help clarify that, I'd recommend adding text above these graphs to distinguish During or After AC stimulation.

Figure 6. I seem to remember that null mutations in *mec-10* can sense touch because with it missing, MEC-4 can form homomeric channels. The *mec-10* hypomorphic mutants cannot sense touch. Is *mec-10(tm1552)* defective in touch or not?

Figure 6G. Was this peak speed? When? Maybe explain with text above the graph?

Discussion: Clarify which neuropeptides are effected or not effected by *egl-3* mutation.

Line 296: I thought according to Fig 5F you did see a positive result for *mec-4*.

Line 308: citation is missing

Line 343: I'm interested in seeing how the duration and strength of the electric shock here compares with those used in fear conditioning paradigms in other experimental animal models and humans.

Reviewer #2 (Comments for the Authors (Required)):

The present study by Tee et al., demonstrates that *C. elegans* exhibits a persistent speed increase in response to AC electric stimulation thus reflecting a fear-like state in the animal that has not been reported previously. The authors have nicely demonstrated that food does not influence the emotional state of behavior through well designed experiments. This study also identifies certain genes that are required to elicit this response. While I believe this study is well presented and may evoke strong interest in the field, I do have some minor concerns regarding certain experiments and would also recommend one key experiment be added before its publication.

1. In Figure 1 supplemental video 3 and 4, the animals shown when electrically stimulated do not show any movement as if a state of shock. However, the authors claim this is not the case for every plate. I would suggest showing a representative video and explain why animals show such varied response. Is the voltage or frequency not uniform in the plate?

2. The authors should address why would DC and AC current have totally different response in the animal. Can the animal sense difference between DC and AC currents as one would expect the animal to just respond to electric stimulation?
3. I would suggest moving Figure 1 that shows the set up to supplemental figure and moving Supplemental Figure 2 that highlights on and off response to the main figures.
4. Since fleeing response is mostly forward locomotion and bends, it wasn't clear in the list of genes analyzed, if the genes involved in this locomotory behavior were tested.
5. To gain insight into how persistent behavior is generated, it is important to demonstrate the key neurons that influence behavior states in the animal. To this effect, the authors have identified a few genes, but do not try to uncover the neurons that might be affected. It wouldn't be hard to express calcium sensor in neurons (for example the motor neurons) and image these during before, during and after electric stimulation to identify which neurons specifically respond during fleeing.

Reviewer #3 (Comments for the Authors (Required)):

Tee et al. present a novel experimental paradigm for persistent behavior in *C. elegans* induced by transient electric shocks with AC stimulation. This behavior is modulated by voltage, duration and frequency of the electric stimulus, showing an ON response at a lower voltage (30 V) and an OFF response at higher voltage (75 V). They demonstrate that the behavior shows persistency, that is, it lasts for up to several minutes after a short stimulus; scalability - meaning that it varies in duration with different intensity of the stimulus; and that it has negative valence, where the response to the presentation of the electric shock overrides foraging behavior. The authors also show that the avoidance response depends on the EGL-19 and UNC-2 voltage-gated calcium channels, as well as the SLO-1 BK channel - but is not dependent on monoamine signaling. Instead, its persistency is modulated by neuropeptide signaling.

Interestingly, these avoidance responses to AC stimuli appears to be different from electrosensory behavior to DC stimuli published by the Samuel lab in 2007 - for example, DC responses are dependent on five amphid neurons, whereas mutants that eliminate amphid function still display the AC responses.

Persistent behavior and concomitant persistency in brain states represent the different brain states found in humans and other animals, such as motivation or emotion. The behavioral paradigm presented entails three of four hallmarks of emotion that were proposed by Anderson and Adolphs, who defined these characteristics specifically with the study of animal models of emotion in mind.

Because of the importance of persistent behavioral states for an understanding of the function of the brain, it is highly valuable that this study establishes a new paradigm in the *C. elegans* genetic model. Electric shocks are extensively used in the study of associative learning in other models, especially rodents, and a comparable *C. elegans* paradigm can be useful for the study of conserved principles of cognition across species. The establishment of the paradigm opens up the possibility to use electric shock as an unconditioned stimulus in experiments on associative learning in *C. elegans*. In this regard, I note that emotional processing e.g. in the amygdala is modulated by monoamine signaling, especially dopamine and serotonin. It may be possible to uncover a role of these modulators in AC electrosensation in future studies.

Since the study explicitly interpret their finding in the light of the concept of four hallmarks of emotion, it would significantly benefit this study - and strengthen the argument made by the authors - to also address the fourth hallmark, generalization of emotional states. This could be done for example by testing if responses to a sensory stimulus of a different modality, applied shortly afterwards, are altered by a previously applied electric shock - such as a blue light stimulus. A different way of address the question of generalization would be to test if the emotional response demonstrates pleiotropy: that is, if there are consequences of an emotion state that "fan out" to a multitude of effects in response to the sensory stimulation, such as vegetative effects. Examples in *C. elegans* would be to measure if pharyngeal pumping or defecation frequency also change after a short electric shock.

Overall, the experiments have been carefully executed and appropriately interpreted.

Importantly, the authors address two caveats - whether the increase in speed is due to food leaving, induced by the electric stimulus; and whether it is due to an increase in temperature. To further address this caveat, it would be desirable to measure the temperature on the surface of the agar plate with the electric stimulus regime applied here. Thirdly, the OFF behavior shown after 75 V stimuli, where no or little increase in speed occurs during the stimulation, could simply be an inhibition of locomotion by interference with neural communication due to the externally applied strong electric field. The authors provide several arguments however that this is not the case - for instance, that a reduction of salt to decrease the current does not change the OFF behavior. Instead, because it occurs at a particular high voltage, it could be an example of a "freeze" response to a strong shock; this can be seen in *C. elegans* as well in response to particularly harsh touch.

In the discussion, I miss a discussion of how this paradigm is actually also relevant for arousal, which is a concept closely related to emotion. Indeed, the hallmarks of persistency, scalability and valence would seem to apply to arousal states as well.

Specific points:

- At least to demonstrate a causative role of *egl-19* in AC responses, a transgenic rescue of the gene should be tested if it restores the avoidance behavior. Such a genomic rescue strain has been published by Gao and Zhen (10.1073/pnas.1012346108). Notably, in this study they used the strain to generate mosaic animals that express *egl-19* only in neurons but not muscles. Data from such a neuron-specific rescue would provide further support for the hypothesis that EGL-19 may act as part of an electrosensory mechanism in neurons.
- Methods section: A description of how the stripes of food were seeded is not provided.
- Statistics: The authors are performing the Kruskal-Wallis test on repeated measures for figures 2 and 3 (the ones with speed measurements before, during and after electric stimulation - assuming that it is the same animals from which three measurements are taken at different time points based on the description), which is not appropriate for these data because Kruskal-Wallis requires the samples to be independent. A Friedman test (also non-parametric, but designed for repeated measures/related samples) would be adequate instead. For the other figures (5 and 6), the Kruskal-Wallis tests used are appropriate, as here they are used to compare between different genotypes.
- It is not mentioned if any pre-processing of the data was done to check for the shape of their distributions or the variance sizes. Speed data often show normal distributions, which would allow to employ a parametric test instead.

In conclusion, I support acceptance of this study for publication, once these points have been considered

Re: "Electric shock causes a fear-like persistent behavioral response in the nematode *C. elegans*" by Tee et al; GENETICS-2022-305222.

Author response

We are most grateful to the editors and the reviewers for the constructive and insightful comments on our original manuscript. We would like to apologize for the fact that this revision took a very long time, which I will explain first below.

Associate Editor, Prof. Arantza Barrios, suggested the following:

I am pleased to inform you that, with minor revisions, it is potentially suitable for publication in GENETICS. The reviewers have comments and concerns that need to be addressed in a revised manuscript.

It is most important that you address the following in your resubmission. The evidence for the behavioural response to AC being a primitive for fear is not strong enough yet. It is unclear whether the behavioural response has negative valence and whether it can be generalised (two of the characteristics extracted by Anderson and Adolphs). One way to address both of these is to test whether the AC treatment can negatively condition a stimulus (salt, benzaldehyde or butane would be good candidates) in a simple associative learning task. In addition, some knowledge of the neurons involved in sensing or executing the behavioural responses to AC would highly improve the manuscript. There are some suggestions about how to do this by the reviewers below.

According to the suggestions, we conducted two things: (1) Test the possibility of the electric shock working as an unconditioned stimulus, and (2) *egl-19* RNAi to see its site-of-action.

(1) Testing the possibility of the electric shock working as an unconditioned stimulus

Conditioned stimulus could be, for example, salt, benzaldehyde, butanone, or temperature. We considered that a salt (NaCl) stimulus would be the most stable during repeated turn ON/OFF of electric shock compared to odorant or temperature stimulus because odor would diffuse and temperature would increase during repetitive conditioning with electric shocks. We have learned all the details of the salt conditioning paradigm from the lino laboratory, who established the method, and tried as many different conditions as possible, such as the presence or absence of bacterial food, different salt concentrations during training, the durations and amplitudes of voltage stimulation, and different interval of the stimulations, etc. However, we have not been able to find a condition in which animals with electric shock significantly move in an opposite direction from animals without shock (the best result we obtained is shown in Fig. 1 below). Based on this result, we decided to tone down our argument that the persistent response represents the "emotion" of worms, as suggested by Reviewer #1, from the Title, the Abstract (lines 28 and 31), and lines 65, 275-276, 412-413, and 424.

Fig. I, Result of conditioning with salt and electric shock. About 50 – 150 animals were placed on a 6 cm NGM plate (25 mM NaCl) seeded with OP50 and provided with 15 V of electric shock for 2 min, followed by a 1 min no voltage interval, which was repeated 10 times. The animals were then collected with KP25(50) wash buffer (50 mM NaCl, 1 mM MgSO₄, 1 mM CaCl₂, 25 mM potassium phosphate, 0.5 g/L gelatin) and transferred to a 9 cm plate with a background concentration of 50 mM NaCl and spots of 0 mM and 150 mM NaCl at opposite ends of the plate. “Chemotaxis index = -1 / +1” reflects the proportion of animals that moved to the 0 mM or 150 mM spot, respectively. The mean position of worms on one plate is represented as a dot, and the results of 10 plates are shown. $p = 0.075$.

(2) *egl-19* RNAi to see its site-of-action

Because full-length *egl-19* cDNA has not been reported so far at least to our knowledge, we first tried feeding RNAi using RNAi-enhanced strains, including the cell-type selective ones as suggested by Reviewer #1. Feeding RNAi in non-neuronal cells (for example, with *gon-1* for germline and *unc-22* for muscles; data not shown) worked well, and the RNAi-sensitive *lin-15B; eri-1* strain exhibited significant suppression of the electric shock-evoked speed increase in an *egl-19(RNAi)*-dependent manner (Figure 5D and E; lines 215-223). It should be noted, however, that the speed increase (*i.e.* Δ speed) in the strain was substantially lower compared to the wild-type animals even without *egl-19(RNAi)* (~0.2 vs ~0.08 mm / s in the wild-type and *lin-15B; eri-1* animals, respectively: Figure 5D and E, scatter plots), probably because of the mutations. We then tested cell type-selective RNAi, but we did not observe significant effects of *egl-19(RNAi)* in any of Glu /ACh/GABA-selective RNAi strains (Fig. II below). Thus, we were not able to identify the neuron type(s), in which *egl-19* functions. We would also like to note that we were not able to reproduce the cell type-selective behavioral defects caused by *unc-13(RNAi)* or *eat-4(RNAi)* as reported in the original paper (Firnhaber and Hammarlund, 2013), possibly suggesting that cell-selective RNAi may not have worked effectively under our experimental conditions, despite our multiple attempts at optimization.

Fig. II, Results of cell type-selective feeding RNAi. *egl-19* feeding RNAi was conducted on XE1582 (Glu), XE1581 (ACh), and XE1375 (GABA) as described in Figure 4E, and no statistically significant difference was observed.

Still, to confirm that EGL-19 functions in neurons, not in muscles, we conducted the injection of *egl-19* double-stranded RNA under the pan-neuronal *rab-3* promoter and observed significant suppression of the electric shock-evoked fleeing response (Figure 5F, lines 223-226). Moreover, we also used a muscle-specific rescue strain of *egl-19* (Gao and Zhen, PNAS 2011, suggested by Reviewer #3) and observed no rescue of the response (Figure 5G, lines 226-229). All of these results indicate that *egl-19* functions in neurons, not in muscles (see also lines 328-330 in Discussion).

Because the number of panels in Figure 5 (ON responses of mutants) increased, we created a new Figure 7 and moved the ON and OFF results of neuromodulator (bioamines and neuropeptides) mutants in Figures 5 and 6 to Figure 7.

(The cloning of *egl-19* gene was exceptionally challenging. Previously, we tried to clone its full-length cDNA but only obtained partial fragments with multiple mutations. In the current study, the bacteria carrying the most effective *egl-19* "Fragment 3" plasmid was very difficult to grow, and we have been unable to isolate the *rab-3p::egl-19* (*Fragment 3, sense*) plasmid for the injection despite months of trials. Fortunately, the PCR fusion method came to our notice and solved the problem.)

Again, we would like to apologize that the results were not so positive despite such a long period of revision. However, we did our best during this time.

Detailed point-to-point responses are listed below.

Reviewer #1:

Major issues:

1. It would have been interesting to have determined which sensory neurons, if any, are required for these AC responses; however, they found that many sensory pathways appear to be dispensable for this response. This raises the possibility that multiple sensory pathways sense AC redundantly in parallel, and they missed this result because they didn't break the right combination of sensory neurons. This needs to be stated in the Discussion.

Response:

Thank you for the suggestion. We described the possibility in the Discussion part (lines 72, 320-321).

*Alternatively, sensory neurons might not even be required to respond. Indeed, the calcium and potassium channels that are required for response are expressed throughout the nervous system. I wonder if they could use cell-selective expression of RNAi to knock-down *egl-19* in at least a large subset of sensory neurons to deduce whether sensory neurons are in fact required for this response? Or use RNAi by feeding in a worm strain that expresses *sid-1* in sensory neurons? This would be very*

interesting. Some useful public strains include: TU3403 (touch neurons), XE1375 (GABA), XE1474 (dopamine), XE1581 (ACh), and XE1582 (glutamatergic).

Response:

We appreciate the reviewer's suggestion, and indeed conducted the experiments. However, as described above in the response to Editor, none of the cell type-selective enhanced RNAi strains exhibited a significant difference when fed *egl-19* interfering RNA (Fig. II), although the RNAi-sensitive *lin-15B; eri-1* strain exhibited significant suppression of the response (Figure 5D and E, lines 215-223). Based on these results, we toned down the possibility of *egl-19* being the sensor by deleting the corresponding sentence (after line 212).

As for the involvement of *egl-19* in neurons, we confirmed it by the injection of *rab-3p::egl-19* dsRNA (Figure 5F and lines 223-226). We are currently narrowing down the cells in which *egl-19* functions using different types of promoters, which we would like to report in our next paper.

2. I cannot tell from your experiments if the heightened speed during or after shock reflects "fear" or other types of emotion. Do any of the AC stimuli serve as effective unconditioned cues for associative learning? This can be easily tested. Without testing this, I find it hard to know if the worms are "excited", "happy", or "fearful" when maintaining their speed during AC or moving fast afterwards. Why not test whether worms form a negative association with innately attractive NaCl in the agar substrate with a chemotaxis assay after training. Then you might have clear results. There are other useful ways to test for negative association too. Alternatively, I would be happy if they just toned down the idea of fear without more experimental evidence.

Response:

Thank you for the suggestion. We have conducted chemotaxis assays in order to test negative association as suggested. As described above in the response to Editor, we did our best to identify a condition in which electric shock worked as an unconditioned stimulus, but we were not able to do so (Fig. I). Therefore we decided to tone down the idea of fear as the reviewer suggested (the Title, lines 28, 31, 65, 275-276, 412-413, and 424).

Minor issues:

Line 67. "express" probably should be rewritten as "reflect"

Response:

Thank you for the suggestion. We rewrote the term as suggested (lines 70 and 31).

Line 68. ...the response if not mediated by any single well-known... (you need to say this because they may be mediated by more than one in redundantly in parallel)

Response:

Thank you for the suggestion. We altered the sentence as suggested (line 72).

Line 73. ...indicate that the response of C. elegans to electric shock...

Response:

Thank you for the suggestion. We changed the sentence as suggested (line 76).

Line 83. Please add info here and elsewhere in Results and Discussion from Chrisman et al., 2016. They got distinct results from Gabel et al. 2007

Response:

Thank you for pointing this out. We have added the information from Chrisman et al. 2016 in Results (lines 86-87) and in Discussion (lines 311-312).

*Line 100. Was there any mutant that speeds up *during* this stimuli?*

Response:

As far as we investigated, we did not discover mutant that speeds up during the electric stimulation.

Lines 109-113. This is confusing as written. I'd recommend taking out this out here and elaborate more carefully in the Discussion.

Response:

Thank you for the comment. We removed the sentence to avoid confusion.

Figure 2A and B. The think lines are supposed to reflect individual measurements and not SEM. Then why does absolute speed measurements appear to drop below zero? This doesn't make sense.

Response:

Once again, we apologize for any confusion caused. The resolution of our tracking system is 0.02 mm/s, and the minimum values were mostly 0.02 mm/s under the 30 V condition, occasionally dropping to 0 mm/s under the 75 V condition. To improve data representation, we revised the figures (now labeled as Figure 1B and 1C): In the new graphs, the vertical axes start from zero.

Figure 2C. I'm confused why you needed to have worms start on the food. Why not have the worms start anywhere. The presence of the food seems to complicate interpretation of results because you do not clarify in most cases when worms are on or off food.

Response:

Thank you for your question. This is because of technical reasons. To process as many plates as possible per day, we need a certain period of interval (1–3 hours generally) from transferring worms to the assay plate until the beginning of the assay. By making a small food patch in the center of the plate, most of the worms locate at the food patch, which made us easily track the worms for longer

periods of time efficiently; if we place worms on no food plate, the worms spread over the entire plate during the interval and sometimes hit the wall soon and become unable to track. We believe that we clearly showed that the presence or absence of food does not affect the persistent speed increase in Figure 4D and E and Sup. Figure 4, and lines 157-176. For clarity, we added a new Sup. Videos 5 and 6 of worms' response on a three-stripe food plate.

Figure 4A. I'd recommend adding sublabels to help explain the six results you have within this single panel. For instance, the bottom one where you stimulate for 600 seconds is interesting. You speculate that the worms may be experiencing exhaustion. To determine this, you might try using laser blue light to stimulate the worms. This may dishabituate the worms and cause them to speed up again, or not.

Response:

Thank you for your recommendation on adding sublabels. We have added sublabels in Figure 4A (30 V, continuous stimulation) and in 4C (75 V, continuous stimulation) for better presentation of the data.

As for the suggestion of an exhaustion experiment, we consider that the experiment of intermittent stimulation addresses the possibility of worms experiencing exhaustion (Figure 4B). In this experiment, the worms were able to maintain a speed increase for a much longer time (compare with Figure 4A, 10 min), suggesting that the worms might experience sensory adaptation instead of motor fatigue during a long stimulation (lines 138-144).

Figure 5. You could try backing up your egl-19 results with L-type calcium channel blocker that effects C. elegans. This would test if you get the same result even if you transiently block function of the channel rather than lack EGL-19 channel the entire life of the animal.

Response:

We agree that the use of a blocker would provide additional support for our results. However, we have opted not to include this experiment in the current version for two primary reasons: (1) Multiple previous studies have demonstrated similar behavioral outcomes between *egl-19* mutants and wild-type worms treated with the blocker (Nemadipine A). (2) The suggested experiment would necessitate including the drug in agar plates. This would require a substantial quantity of the drug, leading to a significant increase in cost, which has made us hesitant to conduct this experiment at the present time.

Figure 5 E and F. You are measuring different time windows in these two panels. To help clarify that, I'd recommend adding text above these graphs to distinguish During or After AC stimulation.

Response:

Thank you for pointing this out. We have separated these panels into different figures and added sublabels (now Figure 5C and Figure 7D).

Figure 6. I seem to remember that null mutations in mec-10 can sense touch because with it missing, MEC-4 can form homomeric channels. The mec-10 hypomorphic mutants cannot sense touch. Is mec-10(tm1552) defective in touch or not?

Response:

Indeed, the Mec phenotype is weaker in *mec-10(tm1552)* compared to the phenotype in *mec-4*. We have changed the corresponding sentence (lines 196-197).

Figure 6G. Was this peak speed? When? Maybe explain with text above the graph?

Response:

This speed is the speed during the first 30 seconds of electric stimulation. We have added sublabel above the graph for better explanation (now Figure 6D).

Discussion: Clarify which neuropeptides are effected or not effected by egl-3 mutation.

Response:

egl-3 mutation affects biosynthesis of many FMRFamide-like peptide (FLP) and neuropeptide-like protein (NLP), but not insulin-like peptide (Husson et al., Prog Neurobiol 2007). We have added this information in Discussion (lines 363-366).

Line 296: I thought according to Figure 5F you did see a positive result for mec-4.

Response:

Sorry for the mistake. We have corrected the sentence properly (lines 319-320).

Line 308: citation is missing

Response:

Thank you for pointing out. We cited the corresponding our result (line 333)

Line 343: I'm interested in seeing how the duration and strength of the electric shock here compares with those used in fear conditioning paradigms in other experimental animal models and humans.

Response:

Thank you for your interest and suggestion. We have added some information on the duration and strength of the electric shock used in fear conditioning paradigms on rodents in lines 370-372.

Reviewer #2:

1. In Figure 1 supplemental video 3 and 4, the animals shown when electrically stimulated do not show any movement as if a state of shock. However, the authors claim this is not the case for every plate. I would suggest showing a representative video and explain why animals show such varied response. Is the voltage or frequency not uniform in the plate?

Response:

Thank you for the comment. We replaced the videos 3 and 4 with the ones showing 2 different responses (worms show no movement or some movement) when the worms are electrically stimulated with 75 V for OFF response. We consider that the voltage and frequency should be uniform in the plate as we used a wide copper plate that covered almost every part of the agar (Figure 1A and D), so the electricity should run between two copper plates evenly. We speculate that a stronger electric stimulus causes the freezing-like response of worms with a certain threshold value (line 109-110), although its details should be further investigated in the future.

2. The authors should address why would DC and AC current have totally different response in the animal. Can the animal sense difference between DC and AC currents as one would expect the animal to just respond to electric stimulation?

Response:

Thank you for the question and suggestion. Our results of *che-2* mutation (affecting most of the amphid neurons; Fujiwara et al., Development 1999) indicate that the amphid sensory neurons required for the DC response, such as ASH, ASJ, and AWC, are not substantially responsible for the response to AC stimulus, at least not solely (Figure 5A, C, Figure 6A and C). This may be the reason why DC and AC stimuli cause totally different responses (lines 311-314).

3. I would suggest moving Figure 1 that shows the set up to supplemental figure and moving Supplemental Figure 2 that highlights on and off response to the main figures.

Response:

We appreciate your suggestion. We have combined Figure 1 with Figure 2 (now labeled as Figure 1), and moved Supplemental Figure 2 as Figure 2.

4. Since fleeing response is mostly forward locomotion and bends, it wasn't clear in the list of genes analyzed, if the genes involved in this locomotory behavior were tested.

Response:

Thank you for pointing this out. We did not analyze genes related to forward locomotion and bends because in this study we focused on the mechanism of sensation (sensory genes) and of persistency (neuromodulators). We are indeed planning to identify neurons more related to locomotion as a next step.

5. To gain insight into how persistent behavior is generated, it is important to demonstrate the key neurons that influence behavior states in the animal. To this effect, the authors have identified a few genes, but do not try to uncover the neurons that might be affected. It wouldn't be hard to express calcium sensor in neurons (for example the motor neurons) and image these during before, during and after electric stimulation to identify which neurons specifically respond during fleeing.

Response:

Thank you for your suggestions. While we agree that identifying the involved neurons would provide significant insights, we believe such details are best reported in a subsequent paper. This is primarily because, even if we detect calcium responses correlating to the behavioral response in specific neurons, confirming causality would still require us to activate or inactivate these neurons. We believe our current data set is well-suited for this initial publication on the behavioral response with genetic analysis, and we intend to deliver a comprehensive analysis of the relevant neural activities in the response in our follow-up study.

Reviewer #3:

*Since the study explicitly interpret their finding in the light of the concept of four hallmarks of emotion, it would significantly benefit this study – and strengthen the argument made by the authors - to also address the fourth hallmark, generalization of emotional states. This could be done for example by testing if responses to a sensory stimulus of a different modality, applied shortly afterwards, are altered by a previously applied electric shock – such as a blue light stimulus. A different way of address the question of generalization would be to test if the emotional response demonstrates pleiotropy: that is, if there are consequences of an emotion state that “fan out” to a multitude of effects in response to the sensory stimulation, such as vegetative effects. Examples in *C. elegans* would be to measure if pharyngeal pumping or defecation frequency also change after a short electric shock.*

Response:

We appreciate your suggestion. As addressed in our previous responses to the Editor and Reviewer #1, we tested the possibility of the electric shock functioning as an unconditioned stimulus. Unfortunately, we did not find a condition for significant difference (as shown in Fig. 1).

Overall, the experiments have been carefully executed and appropriately interpreted. Importantly, the authors address two caveats – whether the increase in speed is due to food leaving, induced by the electric stimulus; and whether it is due to an increase in temperature. To further address this caveat, it would be desirable to measure the temperature on the surface of the agar plate with the electric stimulus regime applied here.

Response:

In response to your suggestion, we measured the temperature change of the agar with the application of the 30 V stimulus for 5 seconds, which induces a robust response. We found no discernible difference in temperature (lines 206-207, 318-319, and 466-471).

Thirdly, the OFF behavior shown after 75 V stimuli, where no or little increase in speed occurs during the stimulation, could simply be an inhibition of locomotion by interference with neural communication due to the externally applied strong electric field. The authors provide several arguments however that this is not the case – for instance, that a reduction of salt to decrease the current does not change the OFF behavior. Instead, because it occurs at a particularly high voltage, it could be an example of a “freeze” response to a strong shock; this can be seen in C. elegans as well in response to particularly harsh touch.

Response:

We appreciate and agree with the comment. However, we have been unable to find out literature detailing the specifics of the "freezing" phenomenon in worms. There are several pieces of literature about seizure events in worms (Lisley et al., PLoS ONE 2016; Lisley et al., Invert. Neurosci. 2018), but these typically require approximately 30 seconds for recovery and do not result in a speed increase.

In the discussion, I miss a discussion of how this paradigm is actually also relevant for arousal, which is a concept closely related to emotion. Indeed, the hallmarks of persistency, scalability and valence would seem to apply to arousal states as well.

Response:

Thank you for raising this intriguing point. According to "dimensional theories" of emotion, emotions are often represented along axes of valence and arousal (for example, Russell, Psychol. Rev, 2003).

Specific points:

At least to demonstrate a causative role of egl-19 in AC responses, a transgenic rescue of the gene should be tested if it restores the avoidance behavior. Such a genomic rescue strain has been published by Gao and Zhen (10.1073/pnas.1012346108). Notably, in this study they used the strain to generate mosaic animals that express egl-19 only in neurons but not muscles. Data from such a neuron-specific rescue would provide further support for the hypothesis that EGL-19 may act as part of an electrosensory mechanism in neurons.

Response:

We appreciate the suggestion. However, we consider that using transgenic mosaic animals for behavioral analysis is not practical because it would require a certain amount of effort to obtain a sufficient number of mosaic animals for behavioral analysis (n = 20–30). Instead, we tried feeding RNAi (suggested by Reviewer #1), pan-neuronal expression of *egl-19* dsRNA, and muscle-rescued *egl-19* animals (reported in the Gao and Zhen paper), all of which indicated that *egl-19* functions in neurons but not in muscles (Figure 5D and G).

Methods section: A description of how the stripes of food were seeded is not provided.

Response:

Sorry for the lack of information. We have added the description of how the stripes of food were seeded in Materials and Methods (lines 495-498).

Statistics: The authors are performing the Kruskal-Wallis test on repeated measures for figures 2 and 3 (the ones with speed measurements before, during and after electric stimulation – assuming that it is the same animals from which three measurements are taken at different time points based on the description), which is not appropriate for these data because Kruskal-Wallis requires the samples to be independent. A Friedman test (also non-parametric, but designed for repeated measures/related samples) would be adequate instead. For the other figures (5 and 6), the Kruskal-Wallis tests used are appropriate, as here they are used to compare between different genotypes.

Response:

Thank you for pointing this out. In our understanding, Friedman test can only be used when all the repeated pairs have values. In our case, however, as the stimulation period increased, the number of worms remaining on the plate decreased due to them hitting the plate wall. This caused missing pairs, therefore the Friedman test could not be carried out.

It is not mentioned if any pre-processing of the data was done to check for the shape of their distributions or the variance sizes. Speed data often show normal distributions, which would allow to employ a parametric test instead.

Response:

We have not performed any pre-processing of the data to check for the data distributions. It is because, in our understanding, a parametric test should be used only on data for which normal distribution is guaranteed, and we have not been able to find a literature describing that speed data can be regarded to have a normal distribution.

July 21, 2023

RE: GENETICS-2022-305494

Dear Dr. Kimura:

I am pleased to accept your manuscript entitled "**Electric shock causes a fleeing-like persistent behavioral response in the nematode *Caenorhabditis elegans***" for publication in GENETICS, pending minor revision (as suggested by reviewer 1).

All reviewers and myself empathise with the authors' struggle to address empirically some of the concerns that were initially raised and we are satisfied with your responses. Please submit your revision along with a brief description of how you modified the manuscript in response to the reviewers' concerns and suggestions (which can be viewed at the bottom of this email). Mainly, the revisions should include the couple of citations that reviewer # 1 highlights. I expect you should be able to submit a revised manuscript within 30 days. A suitably revised manuscript will be acceptable for publication; I don't expect to send it out for review.

When revising the ms., please make an effort to shorten it, because that almost always improves a manuscript. We urge authors to heed the advice of Strunk and White: "omit needless words"¹. Follow this link to submit the revised manuscript: Link Not Available

Thank you for submitting this story to Genetics.

Sincerely,

Arantza Barrios
Associate Editor
GENETICS

Approved by:
Oliver Hobert
Senior Editor
GENETICS

Reviewer comments:

Reviewer #1 (Comments for the Authors (Required)):

Tee et al. provide an update on their fun study that examines how *C. elegans* responds to electric shock. They have worked hard to solidify their finding that *egl-19* is required for the response. They also attempted to check if shock conferred a negative association in plasticity paradigm. I'm overall satisfied and happy with their response. I have a few comments below.

I totally respect the lino's lab expertise on associative learning with NaCl as a cue. However, I sense that Tee et al may have performed suboptimal follow-up experiments to test if electric shock might serve as an unconditioned negative (fearful) stimulus. Chemotaxis assays can be set up to test naïve *attraction* to moderate concentrations of salt (e.g. Ward, 1973), but the authors instead show avoidance of higher concentrations of salt (Figure 1 in response). For attraction, the chemotaxis plate usually contains a background concentration of zero with a salt gradient. In this condition, the worms robustly perform positive chemotaxis. If worms are starved on NGM plates containing normal salt concentration, they will perform chemotaxis with a negative or neutral (0) index in these conditions, signifying that they associated the salt with a negative experience (lino's JEB paper). The authors claim that they tested a full range of parameters, but I'm not sure if the authors actually performed this experiment and tested whether shock could substitute for starvation as a negative stimulus. As written, it is hard to know if by "spots" they mean small microliter volume drops that would quickly diffuse, resulting in the reported concentration of salt at the two locations (e.g. I doubt there was zero salt). - Nevertheless, I agree with the authors that simply toning down language on "fear" and "emotion" is appropriate without offering further positive results. So I'm fine with their revised paper.

I appreciate Tee et al's struggle to test cellular loci that underlie the behavioral responses to shock. I agree that RNAi for behavioral phenotypes can be tricky, and as such negative results need to be carefully considered. Their new positive results

using RNAi expressed throughout the nervous system is encouraging first step and sufficient, in my opinion, for this revised paper. I look forward to their next study using whole nervous system imaging.

I would encourage the authors to consider the benefits of trying out the L-type calcium channel antagonist in the future. They could lower amount of drug needed if they decrease the volume of agar in the assay plate or use smaller plates. They might also get away with reusing drug plates.

The authors have a chance to show broader relevance of their study by adding citation to Chiba et al who found that *C. elegans* uses static electricity to jump gaps (PMID: 37348502). Be clear on how this DC stimulus differs from AC stimuli however.

They used a technique first described in this paper, so they should cite it (Esposito et al., 2007) PMID: 17459615.

Reviewer #2 (Comments for the Authors (Required)):

Tee et al., have made significant changes to their manuscript and have addressed all the concerns I had with the previous submission. I am satisfied with the response and I would highly recommend this version of the manuscript for acceptance.

Associate Editor comments:

The authors have gone to great length to address the initial concerns of the reviewers and I am satisfied with the new version of the manuscript. I would like to join reviewer #1 in their advise on how to test the potential negative valence of the electric shock through conditioning in future experiments (not required for the current manuscript). The authors should consider testing the ability that pairing an electric shock with benzaldehyde (on food) has to switch the innate attraction that worms display towards 1% benzaldehyde into repulsion (in similar ways as it has been done pairing benzaldehyde with starvation).

Re: "Electric shock causes a fear-like persistent behavioral response in the nematode *C. elegans*" by Tee et al; GENETICS-2022-305494.

Based on the comments, we have made the following revisions:

- 1) Added the descriptions related to Chiba et al. (lines 314-319, 324, 328-331)
- 2) Also added a citation of Esposito et al. in the Method section. (It was already cited in the Results section.) (line 569)
- 3) Read "Omit needless words" and made our best efforts to shorten the manuscript. (various places throughout the manuscript)

July 27, 2023

RE: GENETICS-2022-305494R1

Prof. Koutarou D. Kimura
Nagoya Shiritsu Daigaku
Graduate School of Science
Yamanohata, Mizuho-ku
Nagoya, N/A 4678501
Japan

Dear Dr. Kimura:

Congratulations! We are delighted to inform you that your manuscript entitled "**Electric shock causes a fleeing-like persistent behavioral response in the nematode *Caenorhabditis elegans***" is acceptable for publication in GENETICS. Many thanks for submitting your research to the journal.

To Proceed to Production:

1. Format your article according to GENETICS style, as discussed at <https://academic.oup.com/genetics/pages/general-instructions>, and upload your final files at <https://genetics.msubmit.net>.
2. Your manuscript will be published as-is (unedited-as submitted, reviewed, and accepted) at the GENETICS website as an Advanced Access article and deposited into PubMed shortly after receipt of source files and the completed license to publish. Please notify sourcefiles@thegsajournals.org if you do not wish to publish your article via Advanced Access.
3. We invite you to submit an original color figure related to your paper for consideration as cover art. Please email your submission to the editorial office or upload it with your final files. You can submit a small-sized image for evaluation, and if selected, the final image must be a TIFF file 2513px wide by 3263px high (8.375 by 10.875 inches; resolution of 600ppi). Please avoid graphs and small type.

If you have any questions or encounter any problems while uploading your accepted manuscript files, please email the editorial office at sourcefiles@thegsajournals.org.

Sincerely,

Arantza Barrios
Associate Editor
GENETICS

Approved by:
Oliver Hobert
Senior Editor
GENETICS

note: Please add jnls.author.support@oup.com and genetics.oup@kwglobal.com (or the domains @oup.com and @kwglobal.com) to your email program's "safe senders" list. You will be contacted by both at various points during the production process.